# Meta-Learning Universal Priors
# Using Non-Injective Change of Variables

**Yilang Zhang**
Department of ECE
University of Minnesota
Minneapolis, MN 55414
zhan7453@umn.edu

**Alireza Sadeghi**
Department of ECE
University of Minnesota
Minneapolis, MN 55414
sadeg012@umn.edu

**Georgios B. Giannakis**
Department of ECE
University of Minnesota
Minneapolis, MN 55414
georgios@umn.edu

## Abstract

Meta-learning empowers data-hungry deep neural networks to rapidly learn from merely a few samples, which is especially appealing to tasks with small datasets. Critical in this context is the *prior knowledge* accumulated from related tasks. Existing meta-learning approaches typically rely on preselected priors, such as a Gaussian probability density function (pdf). The limited expressiveness of such priors however, hinders the enhanced performance of the trained model when dealing with tasks having exceedingly scarce data. Targeting improved expressiveness, this contribution introduces a *data-driven* prior that optimally fits the provided tasks using a novel non-injective change-of-variable (NCoV) model. Unlike preselected prior pdfs with fixed shapes, the advocated NCoV model can effectively approximate a considerably wide range of pdfs. Moreover, compared to conventional change-of-variable models, the introduced NCoV exhibits augmented expressiveness for pdf modeling, especially in high-dimensional spaces. Theoretical analysis underscores the appealing universal approximation capacity of the NCoV model. Numerical experiments conducted on three few-shot learning datasets validate the superiority of data-driven priors over the prespecified ones, showcasing its pronounced effectiveness when dealing with extremely limited data resources.

## 1 Introduction

Advances in deep learning (DL) have boosted the notion of "learning from data" with field-changing performance improvements reported across a wide range of applications [24, 15, 52]. Large-scale DL models with high fitting capacity have documented ability to cope with the "curse of dimensionality" by providing compact low-dimensional representations of high-dimensional data. Nonetheless, these high-capacity models typically require protracted training using massive data records. Humans on the contrary, can perform exceptionally well on tasks such as object recognition or concept comprehension with merely a few samples. How to acquire the learning ability of humans in the DL training processes is thus appealing and imperative for a number of application domains, especially when data are scarce or costly to annotate. Examples of such applications include machine translation [52], medical imaging [31], and robot manipulations [27].

Meta-learning, also referred to as "learning to learn," seeks to gather the *prior knowledge* shared across a set of inter-related tasks, to enable quickly solving an unseen yet related learning task using minimal training samples [10]. This form of higher-level learning effectively extracts domain-generic inductive biases from prior tasks, which can be subsequently transferred to learn a new task even with limited data. This mirrors the capability that humans excel at — leveraging past experiences to rapidly acquire new skills. Meta-learning holds the promise of yielding powerful priors with which

38th Conference on Neural Information Processing Systems (NeurIPS 2024).

DL models can generalize better, require fewer data for training, and adapt more effectively to new tasks in dynamically changing environments.

Conventional approaches to meta-learning have relied on hand-crafted techniques to extract prior knowledge [47, 46]. With the advent of DL and growing volume of data, there has been a paradigm shift from such cumbersome procedures towards more efficient data-driven strategies. In particular, the prior information is encoded in hyperparameters, which are shared across tasks and can be fine-tuned using the validation data of all tasks. Utilizing these informative hyperparameters, task-specific learning can be performed even with limited data. Early attempts adopted a neural network (NN) with its weights serving as the shared hyperparameters [55, 45, 33]. The *task-invariant* NN leverages the shared hyperparameters, and training data per task, to output the *task-specific* model. However, the selection of an appropriate NN architecture is tailored to the choice of the task-specific models. In addition, NNs inherently lack interpretability and robustness due to their "black box" nature.

Unlike NN-based meta-learning, model-agnostic meta-learning (MAML) does not rely on any presumptions about task-specific models [10]. Instead, it relies on an iterative optimizer to learn the task-specific model. The task-invariant prior information is embodied in the initialization of the optimizer, which is shared across tasks. By learning an informative initialization, task-specific learning can rapidly converge to a local minimum within a few iterations. Interestingly, the initialization generated by MAML can be viewed as a learnable mean of an implicit Gaussian prior probability density function (pdf) over the task-specific model parameters [18]. Building on MAML, several optimization-based meta-learning algorithms have been advocated to learn different prior pdfs [29, 37, 25, 1, 58]. In addition, theoretical studies have been carried out to further offer insights into these approaches [13, 39, 8, 9, 61]. Nevertheless, the prior models of most existing meta-learning methods are confined to preselected pdfs, such as the Gaussian one, and thus have limited expressiveness, meaning fitting ability. Consequently, generalizing meta-learning to domains that deal with scarce datasets, and need sophisticated priors, remains a challenging and largely uncharted territory.

To improve the prior expressiveness in meta-learning, this contribution puts forth what we term non-injective change-of-variable (NCoV) model, which enables learning a universal data-driven prior from related tasks. The contribution of the resultant method named MetaNCoV is threefold:

i) Our novel NCoV model is proven capable of mapping a known source pdf to an *arbitrary* target pdf. This markedly enhances the model expressiveness, especially in high-dimensional spaces.

ii) Theoretical analysis is provided to demonstrate that a parametric NCoV can approximate a broad spectrum of pdfs, that in turn enables versatile plug-in priors for meta-learning. Moreover, this parametric NCoV inherently provides a task-invariant initialization, rather nicely eliminating the need for its explicit learning.

iii) Numerical tests on three benchmark few-shot learning datasets corroborate our theoretical analysis, and underscore the superior prior expressiveness of the proposed MetaNCoV method compared to meta-learning approaches with prespecified pdfs.

## 2 Problem setup

Meta-learning relies on task-invariant prior information from a collection of $T$ given tasks (indexed by $t = 1, \ldots, T$), to deal with data-limited settings. For each $t$, there is a dataset $\mathcal{D}_t := \{(\mathbf{x}_t^n, y_t^n)\}_{n=1}^{N_t}$ consisting of $N_t$ (data, label) pairs. The dataset is divided into a training subset $\mathcal{D}_t^{\mathrm{trn}} \subset \mathcal{D}_t$, and a validation subset $\mathcal{D}_t^{\mathrm{val}} := \mathcal{D}_t \setminus \mathcal{D}_t^{\mathrm{trn}}$. In addition, a new task indexed by $\star$ is also provided, with its training set $\mathcal{D}_\star^{\mathrm{trn}}$, and an unannotated test set $\mathcal{D}_\star^{\mathrm{tst}} := \{\mathbf{x}_\star^n\}_{n=1}^{N_\star^{\mathrm{tst}}}$ for which the corresponding labels $\{y_\star^n\}_{n=1}^{N_\star^{\mathrm{tst}}}$ are to be inferred. The major premise of meta-learning is that the aforementioned tasks are related through their underlying data distributions or problem structures. This relationship makes it feasible to employ a unified large-scale model such as a deep NN to fit all tasks, with each task tailored by its specific model parameter $\phi_t \in \mathbb{R}^d$. However, as the cardinality $|\mathcal{D}_t^{\mathrm{trn}}|$ can be much smaller than $d$, directly optimizing $\phi_t$ over $\mathcal{D}_t^{\mathrm{trn}}$ could readily lead to overfitting.

Meta-learning addresses this issue by capitalizing on the relationships among tasks. Specifically, since $T$ is considerably large in meta-learning, a *task-invariant* prior can be extracted to capture knowledge across tasks, thereby facilitating the data-limited per-task training. This nested structure of prior extraction and per-task training lends itself to a *bilevel optimization* problem. The inner-level (task-level) optimizes the per-task parameter $\phi_t$ using $\mathcal{D}_t^{\mathrm{trn}}$, and the prior provided by outer-level,

while the outer-level (meta-level) evaluates the trained $\{\boldsymbol{\phi}_t\}_{t=1}^T$ using $\{\mathcal{D}_t^{\text{val}}\}_{t=1}^T$, and refines the prior parameterized by $\boldsymbol{\theta} \in \mathbb{R}^D$.

The bilevel optimization objective of meta-learning can be expressed as

$$\min_{\boldsymbol{\theta}} \sum_{t=1}^T \mathcal{L}(\boldsymbol{\phi}_t^*(\boldsymbol{\theta}); \mathcal{D}_t^{\text{val}}) \tag{1a}$$

$$\text{s.t. } \boldsymbol{\phi}_t^*(\boldsymbol{\theta}) = \operatorname*{argmin}_{\boldsymbol{\phi}_t} \mathcal{L}(\boldsymbol{\phi}_t; \mathcal{D}_t^{\text{trn}}) + \mathcal{R}(\boldsymbol{\phi}_t; \boldsymbol{\theta}), \ t = 1, \ldots, T \tag{1b}$$

where the loss function $\mathcal{L}$ assesses the fit of a task-specific model to a designated dataset, and the regularizer $\mathcal{R}$ quantifies the impact of task-invariant prior. From the Bayesian viewpoint, $\mathcal{L}(\boldsymbol{\phi}_t; \mathcal{D}_t^{\text{trn}}) = -\log p(\mathbf{y}_t^{\text{trn}} | \boldsymbol{\phi}_t; \mathbf{X}_t^{\text{trn}})$ can be interpreted as the negative log-likelihood (nll), and $\mathcal{R}(\boldsymbol{\phi}_t; \boldsymbol{\theta}) = -\log p(\boldsymbol{\phi}_t; \boldsymbol{\theta})$ is the negative log-prior (nlp), where $\mathbf{X}_t^{\text{trn}}$ denotes the matrix collecting all the data vectors in $\mathcal{D}_t^{\text{trn}}$, and $\mathbf{y}_t^{\text{trn}}$ is the corresponding label vector. Using Bayes' rule, it follows that $\boldsymbol{\phi}_t^* = \operatorname{argmax}_{\boldsymbol{\phi}_t} p(\boldsymbol{\phi}_t | \mathbf{y}_t^{\text{trn}}; \mathbf{X}_t^{\text{trn}}, \boldsymbol{\theta})$ is the maximum a posteriori (MAP) estimator.

Unfortunately, the global optimum $\boldsymbol{\phi}_t^*$ in (1b) is generally unreachable when the postulated model is a nonlinear function of $\boldsymbol{\phi}_t$. Hence, a feasible alternative is to rely on an approximate solver $\hat{\boldsymbol{\phi}}_t \approx \boldsymbol{\phi}_t^*$ obtained by a tractable optimizer. Depending on how the alternative solver is acquired, meta-learning algorithms can be categorized as either NN- or optimization-based ones. The former harnesses an NN optimizer $\hat{\boldsymbol{\phi}}_t = \text{NN}(\mathcal{D}_t^{\text{trn}}; \boldsymbol{\theta})$ to model the training process that maps $\mathcal{D}_t^{\text{trn}}$ to $\hat{\boldsymbol{\phi}}_t$, with the sought prior encoded in the NN's learnable weights $\boldsymbol{\theta}$ [41, 17]. Despite the effectiveness of NN optimizers in fitting complex mappings, it is hard to decipher the learned prior due to their black-box nature. To improve the interpretability and robustness of the approximate solver, optimization-based meta-learning decodes the "tractable optimizer" as a cascade of a few optimization iterations. The prior is captured by the shared hyperparameters of the optimizer. The first effort towards this direction is termed MAML [10], which relies on a $K$-step gradient descent (GD) optimizer

$$\boldsymbol{\phi}_t^{k+1}(\boldsymbol{\theta}) = \boldsymbol{\phi}_t^k(\boldsymbol{\theta}) - \nabla \mathcal{L}(\boldsymbol{\phi}_t^k(\boldsymbol{\theta}); \mathcal{D}_t^{\text{trn}}), \ k = 0, \ldots, K-1 \tag{2}$$

where task-invariant initialization $\boldsymbol{\phi}_t^0 = \boldsymbol{\phi}^0 = \boldsymbol{\theta}$ parameterizes the prior information, and $\hat{\boldsymbol{\phi}}_t = \boldsymbol{\phi}_t^K$ gives the desired approximate solver. Interestingly, despite the absence of an explicit regularization term (that is, $\mathcal{R}(\boldsymbol{\phi}_t; \boldsymbol{\theta}) = 0$), it has been shown that, under second-order Taylor approximation, MAML's GD solver (2) satisfies [18]

$$\hat{\boldsymbol{\phi}}_t(\boldsymbol{\theta}) \approx \boldsymbol{\phi}_t^*(\boldsymbol{\theta}) = \operatorname*{argmin}_{\boldsymbol{\phi}_t} \mathcal{L}(\boldsymbol{\phi}_t; \mathcal{D}_t^{\text{trn}}) + \frac{1}{2}\|\boldsymbol{\phi}_t - \boldsymbol{\phi}^0\|_{\boldsymbol{\Lambda}_t}^2$$

where the precision matrix $\boldsymbol{\Lambda}_t$ is determined by $\alpha$, $K$, and $\nabla^2 \mathcal{L}(\boldsymbol{\phi}^0; \mathcal{D}_t^{\text{trn}})$. This observation indicates MAML's optimizer approximately amounts to an implicit Gaussian prior $p(\boldsymbol{\phi}_t; \boldsymbol{\theta}) \approx \mathcal{N}(\boldsymbol{\phi}_t; \boldsymbol{\phi}^0, \boldsymbol{\Lambda}_t^{-1})$, with the shared initialization $\boldsymbol{\phi}^0 = \boldsymbol{\theta}$ serving as its mean vector.

Building upon MAML, various methods have been investigated to learn different prior pdfs in both implicit and explicit forms. For example, recent advances further render the (per-step) precision matrix learnable by replacing it with a $\boldsymbol{\Lambda}$ that is common across tasks. Letting $\boldsymbol{\theta_\Lambda}$ denote the parameter of $\boldsymbol{\Lambda}$, the prior parameter is thus augmented as $\boldsymbol{\theta} := [\boldsymbol{\phi}^{0\top}, \boldsymbol{\theta_\Lambda}^\top]$, where $^\top$ denotes transposition. However, a complete parametrization of $\boldsymbol{\Lambda}$ would result in $\boldsymbol{\theta}$ having prohibitively high dimensionality, that is, $D = \mathcal{O}(d^2)$. To ensure scalability with respect to $D$, $\boldsymbol{\Lambda}$ should have a sufficiently simple structure such as isotropic [39], diagonal [29], and or block diagonal [26, 37] matrices. Inspired by transfer learning, one can instead split the model into an embedding "body" and a classifier/regressor "head," and learn their priors independently; that is, with $\boldsymbol{\phi}_t^{\text{body}}$ and $\boldsymbol{\phi}_t^{\text{head}}$ denoting the corresponding partitions of $\boldsymbol{\phi}_t$, the prior is presumed factorable as $p(\boldsymbol{\phi}_t; \boldsymbol{\theta}) = p(\boldsymbol{\phi}_t^{\text{body}}; \boldsymbol{\theta})p(\boldsymbol{\phi}_t^{\text{head}}; \boldsymbol{\theta})$. On the one hand, the head typically has a nontrivial prior such as the Gaussian one [3, 25]. On the other hand, the body's prior is intentionally restricted to a degenerate pdf $p(\boldsymbol{\phi}_t^{\text{body}}; \boldsymbol{\theta}) := \delta(\boldsymbol{\phi}_t^{\text{body}} - \boldsymbol{\phi}^{\text{body}})$, where $\boldsymbol{\phi}^{\text{body}}$ is a subvector of $\boldsymbol{\theta}$, and $\delta(\cdot)$ is the Dirac delta function. This eliminates the need for optimizing $\boldsymbol{\phi}_t^{\text{body}}$ in (1b), thus markedly lowering the overall complexity for solving (1). Although freezing the body in (1b) allows for escalating the dimension of $\boldsymbol{\phi}_t^{\text{body}}$, it often leads to degraded empirical performance [38] compared to the full update (2). In addition to Gaussian and degenerate pdfs, sparse priors (Laplace distributions) have been investigated in the context of network pruning [50].

# 3 Meta-Learning using non-injective change of variables

Existing meta-learning algorithms rely on a *preselected* pdf to parameterize the prior. However, the chosen pdf can have limited expressiveness; that is, it may have insufficient ability to offer an accurate fit due to its prefixed shape. Consider for instance a Gaussian prior pdf, which is inherently unimodal, symmetric, log-concave, and infinitely differentiable by definition. Such a prior may not be well-suited for tasks with multimodal or asymmetric parametric pdfs.

In this work, we propose to learn a *data-driven* prior pdf that optimally fits the given tasks using a novel non-injective change-of-variable (NCoV) model. We thus term the proposed method as Meta-learning with NCoV (MetaNCoV). In contrast to preselected prior pdfs with fixed shapes, the advocated prior model can dynamically adjust its form to approximate a considerably wide range of pdfs, as will be demonstrated both theoretically and numerically. Furthermore, compared to conventional change-of-variable models such as generative adversarial networks (GANs) [16] and normalizing flows (NFs) [43], the introduced NCoV exhibits enhanced capacity for pdf estimation, especially in high-dimensional spaces. Change-of-variable models and their applications in pdf estimations will be first elaborated. All the proofs are delegated to the Appendix.

## 3.1 Pdf estimation via change of variables

The key idea of change-of-variable model is to identify a transformation $f$, through which a known pdf $p_{\mathbf{Z}}$ can be altered to approximate a target pdf $q$. For instance, GANs [16] and variational autoencoders (VAEs) [21] seek a generator/encoder such that high-dimensional pdfs can be acquired from a low-dimensional latent Gaussian pdf $p_{\mathbf{Z}} = \mathcal{N}(\mathbf{0}, \mathbf{I})$. Due to the dimensional discrepancy between signal and latent spaces, these models are typically utilized to estimate signals living on a low-dimensional manifold; e.g., images.

To enhance the model capactiy as well as pdf tractability, NFs were introduced in [43] as a surrogate variational model for posterior inference. Recently, they have been shown also effective in estimating prior pdfs from a set of unannotated samples [6, 14, 7]. The formulation of NFs relies on the well-known change-of-variable formula. Given a continuous random vector $\mathbf{Z} \in \mathbb{R}^d$ with prior pdf $p_{\mathbf{Z}}$, and a bijection $f : \mathbb{R}^d \mapsto \mathbb{R}^d$, then $\mathbf{Z}' := f(\mathbf{Z})$ is also a continuous random vector with analytical pdf

$$p_{\mathbf{Z}'}(\mathbf{z}') = p_{\mathbf{Z}}(f^{-1}(\mathbf{z}')) \left| \det J_{f^{-1}}(\mathbf{z}') \right| = \frac{p_{\mathbf{Z}}(f^{-1}(\mathbf{z}'))}{|\det J_f(\mathbf{z}')|} \text{ (a.e.)} \tag{3}$$

where $J_f(\mathbf{z}')$ denotes the Jacobian of $f$ at $\mathbf{z}' \in \mathbb{R}^d$, $\det$ is the determinant, and $\det J_f \neq 0$ almost everywhere (a.e.) for bijective $f$. To ensure the invertibility of $f$, a prudent choice is to model it as a composition of a sequence of bijective functions $f = f_1 \circ f_2 \circ \ldots \circ f_n$.

In Bayesian inference [43], $q$ is an intractable posterior, and $f$ is optimized to minimize the KL-divergence between $p_{\mathbf{Z}'}$ and $q$, or equivalently, maximize the so-termed evidence lower bound (ELBO). For density estimation [6], the wanted $q$ is an unknown prior pdf, while $f$ is acquired via maximum likelihood training. The obtained $f$ can be leveraged in two important applications: i) probability estimation $p_{\mathbf{Z}'}(\mathbf{v}) \approx q(\mathbf{v})$ for a given sample $\mathbf{v} \sim q$ using (3), and ii) generation of a sample $\mathbf{z}' = f(\mathbf{z}), \mathbf{z} \sim p_{\mathbf{Z}}$ for which $p_{\mathbf{Z}'} \approx q$.

When $d = 1$, the probability integral transform (PIT) suggests that, the optimal $f^* = Q^{-1} \circ P_{\mathbf{Z}}$ leads to precisely $P_{\mathbf{Z}'} = Q$ a.e., where $Q$, $P_{\mathbf{Z}}$ and $P_{\mathbf{Z}'}$ are the cumulative distribution functions (cdfs) corresponding to $q$, $p_{\mathbf{Z}}$ and $p_{\mathbf{Z}'}$, and $q > 0$ a.e. ensures $Q$ is bijective. The resultant cdf $P_{\mathbf{Z}'} = P_{\mathbf{Z}} \circ f^{*-1}$ is a pushforward measure, also notated as $P_{\mathbf{Z}'} = f^*_\# P_{\mathbf{Z}}$. In high-dimensional spaces ($d > 1$) however, the existence of such an $f^*$ may not hold due to the invertibility assumption of $f^*$; see examples in e.g., [22, Section 4]. In fact, it has been shown that NFs are capable of modeling pdfs with a full support; i.e., when $q > 0$ on $\mathbb{R}^d$ [36].

## 3.2 Improved pdf estimation via non-injective change of variable

To improve the fitting capacity of change-of-variable models for generic $q$, especially those in high-dimensional spaces or without full support, the fresh idea of this work is to waive the injectivity assumption on $f$. In doing so, we can generalize the PIT to an arbitrary $q$, as illustrated in the following theorem.

**Theorem 3.1** (Multivariate PIT). *Let $P_{\mathbf{Z}} : \mathbb{R}^d \mapsto [0,1]$ be the cdf of continuous random vector $\mathbf{Z} :=$ $[Z_1, \ldots, Z_d]^\top$ with $\{Z_i\}_{i=1}^d$ mutually independent. For any differentiable a.e. cdf $Q : \mathbb{R}^d \mapsto [0,1]$, there exists a non-decreasing function $f^* : \mathbb{R}^d \mapsto \mathbb{R}^d$ that the random vector $\mathbf{Z}' := f^*(\mathbf{Z})$ has cdf*

$$P_{\mathbf{Z}'} = Q \ (a.e.). \tag{4}$$

*Remark* 3.2 (Choice of source distribution). In the theorem, the prior distribution for the source random vector $\mathbf{Z}$ can be chosen arbitrarily, as if it is continuous and has mutually independent entries. Popular choices include standard Gaussian $\mathcal{N}(\mathbf{0}_d, \mathbf{I}_d)$ and uniform $\mathcal{U}([0,1]^d)$.

*Remark* 3.3 (Comparison with injective NFs). While conventional NFs (3) require $J_f \neq \mathbf{0}$ a.e. to ensure the injectivity of $f$, Theorem 3.1 relaxes this assumption to allows $f$ being non-injective and thus enables $\mathbf{Z}' = f(\mathbf{Z})$ to match an arbitrary target distribution (even discrete one) in a high-dimensional space. It is worth mentioning that the mild assumption on the differentiability of $Q$ is merely used to guarantee the existence of $q$, which can be easily satisfied. However, one limitation of the advocated NCoV is that it generally has no analytical solution for the resultant surrogate pdf

$$p_{\mathbf{Z}'}(\mathbf{z}') = \int_{\mathbb{R}^d} p_{\mathbf{Z}}(\mathbf{z})\delta[\mathbf{z}' - f(\mathbf{z})]d\mathbf{z}. \tag{5}$$

As a remedy, efficient numerical integration can be performed to estimate $p_{\mathbf{Z}'}$ when $d$ is small. Additional comparisons with NFs and optimal transport are deferred to Appendix E.

While Theorem 3.1 suggests the existence of the optimal $f^*$ that incurs the exact match $p_{\mathbf{Z}'} = q$, the expression for such an $f^*$ relies on the sought $q$, which is typically intractable or unknown. Therefore, a feasible alternative is to resort to a tractable parametric $f(\cdot; \boldsymbol{\theta}_f)$, which approximates $f^*$ by learning $\boldsymbol{\theta}_f$ from the provided data. To further compare NCoVs with NFs, we will focus exclusively on Sylvester NF [51] in the following sections, but our analysis can be readily generalized to other transformations; see Remark 3.7. Sylvester NF was introduced in [51] to improve the expressiveness of planar NF [43] by increasing its "width". In particular, Sylvester NF adopts the form

$$f(\mathbf{Z}; \boldsymbol{\theta}_f) := \mathbf{Z} + \mathbf{A}\sigma(\mathbf{B}\mathbf{Z} + \mathbf{c}), \ \mathbf{Z} \in \mathbb{R}^d \tag{6}$$

where $\mathbf{A} \in \mathbb{R}^{d \times m}, \mathbf{B} \in \mathbb{R}^{m \times d}, \mathbf{c} \in \mathbb{R}^m$ are learnable weights with $m$ being the number of hidden neurons (a.k.a. width), $\sigma$ is an entry-wise nonlinear operator, and $\boldsymbol{\theta}_f := [\text{vec}(\mathbf{A})^\top, \text{vec}(\mathbf{B})^\top, \mathbf{c}^\top]^\top$. It can be easily verified that the Sylvester NF boils down to the planar one when $m = 1$. Akin to other NFs, one can also increase the "depth" of the flows by stacking multiple Sylvester NF layers into a chain $f_1 \circ f_2 \circ \ldots \circ f_n$. The next theorem states that, the optimal $f^*$ can be approximated to arbitrary precision using a sufficiently wide one-layer Sylvester NCoV.

**Definition 3.4.** A random vector on $\mathbb{R}^d$ is said to be tail-convergent if i) it has a pdf $p : \mathbb{R}^d \mapsto \mathbb{R}^+ \cup \{0\}$, and ii) for $\forall \epsilon > 0$ there exists a bounded $E \subset \mathbb{R}^d$ for which

$$\int_{\mathbb{R}^d \setminus E} p < \epsilon. \tag{7}$$

**Theorem 3.5** (Universal approximation via Sylvester NCoVs). *Let $P_{\mathbf{Z}}$ denote the cdf of tail-convergent continuous random vector $\mathbf{Z} \in \mathbb{R}^d$ with mutually independent entries, and $Q$ a Lipschitz cdf of a tail-convergent random vector. For any $\epsilon > 0$, there exists cdfs $\tilde{P}, \tilde{Q}$ for which the corresponding pdfs $\tilde{p}, \tilde{q}$ vanishes outside compact sets $E_p, E_q$, and*

$$|P_{\mathbf{Z}}(\mathbf{v}) - \tilde{P}(\mathbf{v})| < \epsilon, \ |Q_{\mathbf{Z}}(\mathbf{v}) - \tilde{Q}(\mathbf{v})| < \epsilon, \ \forall \mathbf{v} \in \mathbb{R}^d. \tag{8}$$

*Moreover, let $E \subseteq E_p$ be any set on which the optimal $f^*$ matching $\tilde{P}_{\mathbf{Z}}$ to $\tilde{Q}$ is injective and right-continuous. There exists a Sylvester NCoV $f$ and a zero-measure set $E_0$, such that*

$$|f(\mathbf{Z}) - f^*(\mathbf{Z})| < \epsilon, \ \forall \mathbf{Z} \in E_p \setminus E_0, \tag{9a}$$
$$|P_{\mathbf{Z}}(\mathbf{z}) - Q \circ f(\mathbf{z})| < \epsilon, \ \forall \mathbf{z} \in E \setminus E_0. \tag{9b}$$

We have shown that when $f^*$ is injective, the cdf of the optimally transformed $\mathbf{Z}' = f^*(\mathbf{Z})$ can be written as a pushforward $Q = P_{\mathbf{Z}'} = P_{\mathbf{Z}} \circ f^{*-1}$. Likewise, this relationship remains valid when restricting $f^*$ to a set $E$ on which $f^*$ is injective. However, since the Sylvester NCoV $f$ may not be injective on $E$, one cannot directly compare $Q$ with $P_{\mathbf{Z}} \circ f^{-1}$. Fortunately, this pushforward can be equivalently written as $P_{\mathbf{Z}}(\mathbf{z}) = Q \circ f^*(\mathbf{z})$, $\forall \mathbf{z} \in E$; see Lemma B.1 in the Appendix. Utilizing this alternative relationship, Theorem 3.5 states that the Sylvester NCoV $f$ not only approximates $f^*$ a.e. on $E_p$, but also results in pushforward approximation $P_{\mathbf{Z}} \approx Q \circ f$ a.e. on $E$.

---

**Algorithm 1** MetaNCoV algorithm

---

**Input:** $\{\mathcal{D}_t\}_{t=1}^T$, step sizes $\alpha$ and $\beta$, batch size $B$, and maximum iterations $K$ and $R$.

**Initialization:** randomly initialize $\boldsymbol{\theta}_f^0$.

**for** $r = 0, \ldots, R - 1$ **do**

    Sample random $\mathcal{T}^r \subset \{1, \ldots, T\}$ of cardinality $B$.

    **for** $t \in \mathcal{T}^r$ **do**

        Initialize $\mathbf{z}_t^0 = \operatorname{argmin}_{\mathbf{z}_t} \mathcal{R}_{\mathbf{Z}}(\mathbf{z}_t)$.

        **for** $k = 0, \ldots, K - 1$ **do**

            Descend $\mathbf{z}_t^{k+1} = \mathbf{z}_t^k - \alpha \nabla_{\mathbf{z}_t^k} [\mathcal{L}(f(\mathbf{z}_t^k; \boldsymbol{\theta}_f^r); \mathcal{D}_t^{\mathrm{trn}}) + \mathcal{R}_{\mathbf{Z}}(\mathbf{z}_t^k)]$.

        **end for**

        Approximate solver $\hat{\mathbf{z}}_t = \mathbf{z}_t^K$.

    **end for**

    Update $\boldsymbol{\theta}_f^{r+1} = \boldsymbol{\theta}_f^r - \beta \frac{1}{B} \sum_{t \in \mathcal{T}^r} \nabla_{\boldsymbol{\theta}_f^r} \mathcal{L}(f(\hat{\mathbf{z}}_t(\boldsymbol{\theta}_f^r); \boldsymbol{\theta}_f^r); \mathcal{D}_t^{\mathrm{val}})$.

**end for**

**Output:** $\hat{\boldsymbol{\theta}}_f = \boldsymbol{\theta}_f^R$.

---

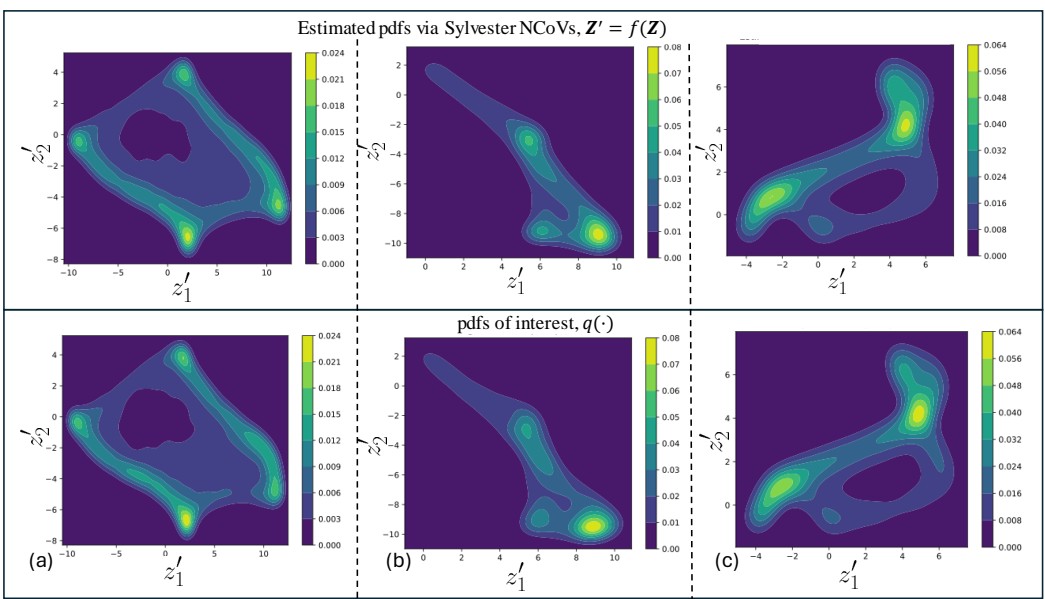

Figure 1: Transforming a standard Gaussian pdf into multi-modal target pdfs using Sylvester NCoVs.

*Remark* 3.6 (Mild assumptions). The assumptions in Theorem 3.5 are mild and common. In particular, tail-convergence only requires the probability of large deviation diminishing to 0 as the norm of the random vector goes to $+\infty$, while imposing no constraint on the decaying rate. This assumption can be easily satisfied by a wide family of distributions, even including the heavily-tailed ones. Under this benign assumption, (8) suggests $P_{\mathbf{Z}}$ and $Q$ can be approximated by alternatives $\tilde{P}, \tilde{Q}$ with pdfs $\tilde{p}, \tilde{q}$ having truncated tails. This is crucial to universal approximation, which typically requires $f^*$ to be bounded or Lebesgue integrable [5]. Moreover, the Lipschitzness of $Q$ is solely utilized to ensure the boundness of its gradient, namely the pdf $q$. This can be also readily met by most practical cdfs.

*Remark* 3.7 (Generalization to other NFs). Although Theorem 3.5 primarily focuses on one-layer Sylvester NCoVs, similar analysis for other transformation $f$ can be acquired by employing different universal approximation models. For instance, results for multi-layer planar NCoVs and multi-layer Sylvester NCoVs can be respectively established leveraging [30] and [32].

*Remark* 3.8 (Influence of $\epsilon$). It is worth noting that the width $m$ of the Sylvester NCoV depends on $\epsilon$ as well as the optimal $f^*$. Smaller $\epsilon$ typically leads to larger $m$. Additionally, the nonlinearity $\sigma$ must be sigmoidal; see Definition B.3 in the Appendix for details.

### 3.3 Meta-learning universal priors via MetaNCoV

Next, we elucidate how universal priors can be learned in meta-learning by harnessing the proposed NCoV model. Different from existing works that rely on prespecified prior forms such as Gaussian pdfs, the novel concept of this work is to learn a data-driven prior that optimally conforms with the given tasks. This is achieved by transforming the random vector $\mathbf{Z} \in \mathbb{R}^d$ with a known prior $p_{\mathbf{Z}}$ to $\mathbf{Z}' = f(\mathbf{Z}; \boldsymbol{\theta}_f)$, whose pdf is given by (5). This $p_{\mathbf{Z}'}$ acts as a surrogate model for the unknown $p(\boldsymbol{\phi}_t; \boldsymbol{\theta})$, and learning the prior parameter $\boldsymbol{\theta}$ thus boils down to optimization of the transformation parameter $\boldsymbol{\theta}_f$. Nevertheless, as discussed in Remark 3.3, $p_{\mathbf{Z}'}$ typically has no close-form expression when $f$ is non-injective. Therefore, instead of directly optimizing $\boldsymbol{\phi}_t$, we propose to optimize the latent vector $\mathbf{z}_t$ corresponding to $\boldsymbol{\phi}_t = f(\mathbf{z}_t; \boldsymbol{\theta}_f)$, which yields

$$\min_{\boldsymbol{\theta}_f} \sum_{t=1}^{T} \mathcal{L}(f(\mathbf{z}_t^*(\boldsymbol{\theta}_f); \boldsymbol{\theta}_f); \mathcal{D}_t^{\text{val}}) \tag{10a}$$

$$\text{s.t. } \mathbf{z}_t^*(\boldsymbol{\theta}_f) = \operatorname*{argmin}_{\mathbf{z}_t} \mathcal{L}(f(\mathbf{z}_t; \boldsymbol{\theta}_f); \mathcal{D}_t^{\text{trn}}) + \mathcal{R}_{\mathbf{Z}}(\mathbf{z}_t), \ \forall t \tag{10b}$$

where $\mathcal{R}_{\mathbf{Z}}(\mathbf{z}_t) := -\log p_{\mathbf{Z}}(\mathbf{z}_t)$ is the nlp regularizer, and $\mathbf{z}_t^*$ is thus the MAP estimator for $\mathbf{z}_t$.

Similar to (2), the global task-level minimizer $\mathbf{z}_t^*$ is generally infeasible to attain. Hence, a tractable alternative is to rely on an approximate GD solver. Interestingly, our formulation (10) naturally offers a convenient initialization using the *maximum a priori estimator*

$$\mathbf{z}_t^0 = \operatorname*{argmax}_{\mathbf{z}_t} p_{\mathbf{Z}}(\mathbf{z}_t) = \operatorname*{argmin}_{\mathbf{z}_t} \mathcal{R}_{\mathbf{Z}}(\mathbf{z}_t), \ \forall t \tag{11}$$

As an example, choosing $p_{\mathbf{Z}} = \mathcal{N}(\mathbf{0}_d, \mathbf{I}_d)$ automatically gives $\mathbf{z}_t^0 = \mathbf{0}_d$ and the corresponding $\boldsymbol{\phi}_t^0 = f(\mathbf{0}_d; \boldsymbol{\theta}_f)$. This elegantly removes the need for separately learning the task-invariant initialization $\boldsymbol{\phi}^0$, which is exactly the maximum a priori estimator of the preselected Gaussian prior pdf $p(\boldsymbol{\theta}_t; \boldsymbol{\theta}) = \mathcal{N}(\boldsymbol{\phi}^0, \boldsymbol{\Lambda}_t)$. In fact, the task-invariant initialization reflects our optimal guess of $\boldsymbol{\phi}_t$ before accessing any task-specific data, and can be naturally derived by maximizing the prior pdf. It also worth noting that while the idea of optimizing the latent (instead of primal) variables shares similarities with [44], the latent space in [44] is designed from a different perspective, which is low-dimensional to the end, and requires an initialization.

To this end, (10) can be solved using a standard alternating optimizer. The resultant MetaNCoV algorithm is listed step-by-step in Algorithm 1, where the inner-level (10b) and outer-level (10a) are respectively optimized using $K$-step GD and mini-batch stochastic GD.

While our idea has the potential to be broadened beyond meta-learning, we must emphasize that our current setup is specifically tailored to meta-learning, which does not require a tractable pdf, but rather demands enhanced prior expressiveness. We should also highlight that the *intractability* of (5) prohibits learning NCoV via conventional approaches such as maximum likelihood training and evidence lower-bound maximization – this thus necessitates careful attention and extra certain designs when applying NCoV to other domains.

## 4 Numerical tests

In this section, we test and showcase the empirical superiority of MetaNCoV on both synthetic and real datasets. All datasets descriptions and hyperparameter setups are deferred to the Appendix C. Codes for reproducing the results are available at `https://github.com/zhangyilang/MetaNCoV`.

### 4.1 Tests with toy data

Here, we investigate an intricate yet interesting scenario to demonstrate the efficacy of NCoVs to approximate complex multi-modal pdfs in two-dimensional (2D) settings. The primary objective is to transform a standard Gaussian random vector $\mathbf{Z} \sim \mathcal{N}(\mathbf{0}_{2 \times 1}, \mathbf{I}_{2 \times 2})$ into multi-modal complex pdfs. The outcomes of this experiment are presented in Figure 1. The lower row displays the ground-truth pdfs $q$ of interest, while the upper row showcases the numerically estimated pdfs of the transformed random vector $\mathbf{Z}' = f(\mathbf{Z})$, where $f$ is a Sylvester NCoV, and the pdf of $\mathbf{Z}'$ is estimated via (5). As clearly evidenced in these results, the advocated NCoVs exhibit their capability to effectively convert

Table 1: Performance comparison of MetaNCoV against meta-learning methods having different priors. For fairness, only methods with a 4-block CNN backbone have been included. The highest accuracy as well as the mean accuracies within its $95\%$ confidence interval are bolded.

| Method | Prior model | 5-class miniImageNet | |
| --- | --- | --- | --- |
| | | 1-shot (%) | 5-shot (%) |
| Meta-LSTM [41] | RNN-based | $43.44_{\pm0.77}$ | $60.60_{\pm0.71}$ |
| MAML [10] | implicit Gaussian | $48.70_{\pm1.84}$ | $63.11_{\pm0.92}$ |
| MetaSGD [29] | diagonal Gaussian | $50.47_{\pm1.87}$ | $64.03_{\pm0.94}$ |
| R2D2 [3] | degenerate body & Gaussian head | $51.8_{\pm0.2}$ | $68.4_{\pm0.2}$ |
| MC [37] | block-diagonal Gaussian | $54.08_{\pm0.93}$ | $67.99_{\pm0.73}$ |
| Warp-MAML [12] | Gaussian | $52.3_{\pm0.8}$ | $68.4_{\pm0.6}$ |
| MAML + L2F [2] | implicit Gaussian | $52.10_{\pm0.50}$ | $69.38_{\pm0.46}$ |
| MeTAL [1] | implicit Gaussian | $52.63_{\pm0.37}$ | $70.52_{\pm0.29}$ |
| Minimax-MAML [58] | inverted Gaussian & entropy | $51.70_{\pm0.42}$ | $68.41_{\pm1.28}$ |
| MAML + MetaNCoV | NCoV-based | $\mathbf{57.74_{\pm1.47}}$ | $70.72_{\pm0.70}$ |
| MetaSGD + MetaNCoV | | $\mathbf{59.10_{\pm1.52}}$ | $\mathbf{71.48_{\pm0.68}}$ |

Table 2: Performance comparison using the WRN-28-10 features [44]. $^\dagger$ indicates that both training and validation tasks are used in the training phase of meta-learning.

| Method | Crop | 5-class miniImageNet | | 5-class tieredImageNet | |
| --- | --- | --- | --- | --- | --- |
| | | 1-shot (%) | 5-shot (%) | 1-shot (%) | 5-shot (%) |
| MetaSGD [29] | | $56.58_{\pm0.21}$ | $68.84_{\pm0.19}$ | $59.75_{\pm0.25}$ | $69.04_{\pm0.22}$ |
| LEO$^\dagger$ [44] | center | $61.76_{\pm0.08}$ | $\mathbf{77.59_{\pm0.12}}$ | $66.33_{\pm0.05}$ | $81.44_{\pm0.09}$ |
| MC [37] | | $61.22_{\pm0.10}$ | $75.92_{\pm0.17}$ | $66.20_{\pm0.10}$ | $82.21_{\pm0.08}$ |
| MC$^\dagger$ [37] | | $61.85_{\pm0.10}$ | $77.02_{\pm0.11}$ | $67.21_{\pm0.10}$ | $82.61_{\pm0.08}$ |
| MetaSGD + MetaNCoV | center | $59.42_{\pm1.32}$ | $70.24_{\pm0.73}$ | $60.36_{\pm1.29}$ | $75.08_{\pm0.66}$ |
| MC + MetaNCoV | | $\mathbf{63.40_{\pm1.30}}$ | $76.12_{\pm0.68}$ | $\mathbf{72.38_{\pm1.26}}$ | $\mathbf{86.47_{\pm0.56}}$ |
| LEO$^\dagger$ [44] | multiview | $63.97_{\pm0.20}$ | $79.49_{\pm0.70}$ | — | — |
| MC$^\dagger$ [37] | | $64.40_{\pm0.10}$ | $80.21_{\pm0.11}$ | — | — |
| MC + MetaNCoV | multiview | $\mathbf{66.54_{\pm1.29}}$ | $\mathbf{86.52_{\pm0.54}}$ | — | — |

a basic Gaussian distribution into intricate multi-modal distributions in 2D. The expressiveness of Sylvester NCoVs and numerical comparison of NFs with NCoVs are postponed to Appendix D.

## 4.2 Performance evaluation using real data

Next, the empirical performance of MetaNCoV is assessed on three real datasets for meta-learning.

The experimental setups follow from the standard $M$-class $N$-shot few-shot classification protocol [41, 10]. In particular, $\mathcal{D}_t^{\text{trn}}$ per task $t$ consists of $M$ randomly drawn classes, each containing $N$ labeled data. The default task-specific model is a standard 4-block convolutional NN (CNN) [55]. Each block of the CNN comprises a $3 \times 3$ convolution layer, a batch normalization layer, a ReLU activation, and a $2 \times 2$ max pooling layer. After the convolutional blocks, a linear regressor with softmax activation is appended to perform classification. Following the practices of [37, 12], the number of convolutional channels is set to 128 to improve its fitting capacity. Additionally, to be consistent with Theorem 3.5, Sylvester NCoVs are adopted in all the tests.

To illustrate the benefit of learning more expressive priors, the first test compares MetaNCoV with other meta-learning algorithms having different prespecified priors using the miniImageNet dataset [55]. As a plug-in prior model, our MetaNCoV can be readily integrated with other meta-learning methods that adopt different task-level optimizers. In this test, we implement MetaNCoV with MAML [10] and MetaSGD [29]. The results are listed in Table 1, where the performance metric is the average classification accuracy with $95\%$ confidence interval on new tasks. It is seen that our MetaNCoV outperforms all the competitors. This empirically confirms the superiority of data-driven priors over the prespecified pdfs, as well as the effectiveness of MetaNCoV in learning an expressive

Table 3: Performance comparison of MetaNCoV against meta-learning and metric-learning methods on the CUB-20-2011 dataset. For fairness, the backbone model is a 4-block CNN.

| Method | Type | 5-class CUB-200-2011 | |
| --- | --- | --- | --- |
| | | 1-shot (%) | 5-shot (%) |
| MatchingNet [55] | metric-learning | $45.30_{\pm 1.03}$ | $59.50_{\pm 1.01}$ |
| MAML [10] | meta-learning | $58.13_{\pm 0.36}$ | $71.51_{\pm 0.30}$ |
| ProtoNet [48] | metric-learning | $37.36_{\pm 1.00}$ | $45.28_{\pm 1.03}$ |
| RelationNet [49] | metric-learning | $58.99_{\pm 0.52}$ | $71.20_{\pm 0.40}$ |
| DN4 [28] | metric-learning | $53.15_{\pm 0.84}$ | $\mathbf{81.90_{\pm 0.60}}$ |
| MattML [62] | meta-learning | $66.29_{\pm 0.56}$ | $80.34_{\pm 0.30}$ |
| MAML + MetaNCoV | meta-learning | $\mathbf{69.24_{\pm 1.36}}$ | $80.41_{\pm 0.60}$ |
| MetaSGD + MetaNCoV | | $\mathbf{69.94_{\pm 1.34}}$ | $80.54_{\pm 0.59}$ |

prior. Moreover, a remarkable performance gain can be observed on the 1-shot dataset. This justifies the claim that prior can be particularly informative when the training data are extremely scarce. For an apples-to-apples comparison, methods that use pre-trained feature extractors or more complicated models (e.g., residual networks) are not included in this table. The compatibility of MetaNCoV to these models will be demonstrated in the subsequent tests.

The second test evaluates MetaNCoV on miniImageNet and tieredImageNet feature embeddings extracted using a pre-trained Wide ResNet(WRN)-28-10 backbone [44]. Compared to the 4-block CNN, this model has a greater number of parameters and thus enhanced expressiveness. The results are summarized in Table 2, where MetaNCoV is implemented with MetaSGD [29] and MC [37]. In all tests, MetaNCoV brings about notable performance improvement compared to the corresponding baselines. This validates MetaNCoV's effectiveness and flexibility as a plug-in prior module.

The last test assesses the performance of MetaNCoV on the CUB-200-2011 dataset [57]. In contrast to the previous two datasets that contain nature images of distinct objects, this dataset specifically focuses on birds of various species. While the classification of nature objects primarily relies on low-level features such as shapes and colors, classifying various birds requires further recognition of high-level features including textures and segmentations. To learn these complicated features, the model needs to be either trained with sufficient data, or equipped with a powerful prior. Table 3 showcases the performances of different meta- and metric-learning methods on such a dataset. Again, our MetaNCoV method is markedly effective on the 1-shot dataset where data are exceptionally limited. This highlights the significance of an expressive prior. For the 5-shot dataset where data are relatively abundant, its performance is also comparable to the state-of-the-art ones.

## 4.3 Ablation study

Next, ablation tests are conducted to analyze the performance gain of MetaNCoV. The test is carried out on the miniImageNet dataset, with results gathered in Table 4. The first ablation investigates the impact of the advocated NCoVs over the injective ones. To ensure the injectivity of the Sylvester NF $f$, we follow the QR parameterization recommended in [51]. One can

Table 4: Ablation tests for MetaNCoV.

| Ablation setup | 5-class miniImageNet | |
| --- | --- | --- |
| | 1-shot (%) | 5-shot (%) |
| NCoV (baseline) | $\mathbf{59.10_{\pm 1.52}}$ | $\mathbf{71.48_{\pm 0.68}}$ |
| Injective NF | $56.72_{\pm 1.46}$ | $69.41_{\pm 0.68}$ |
| ReLU $\sigma$ | $56.54_{\pm 1.46}$ | $69.84_{\pm 0.68}$ |

see the improved performance of NCoV due to its enhanced expressiveness, which numerically verifies Theorem 3.1 and Remark 3.3. The second ablation examines the influence of nonlinear function $\sigma$ in the Sylvester NCoVs. By changing the $\sigma$ from sigmoid to the popular ReLU activation, a degradation of empirical performance can be observed. This observation corroborates with Remark 3.8. Additional experiments and visualizations can be found in Appendix D.

## 4.4 Cross-domain generalization

This subsection showcases the generalization capacity of MetaNCoV in cross-domain few-shot learning. This test is more challenging compared to previous ones due to the domain gap between

Table 5: Performance comparison of MetaNCoV against meta-learning algorithms in cross-domain few-shot learning setups. The prior models are trained on miniImageNet and tested on three datasets.

| Method | 5-class TieredImageNet | | 5-class CUB | | 5-class Cars | |
|---|---|---|---|---|---|---|
| | 1-shot (%) | 5-shot (%) | 1-shot (%) | 5-shot (%) | 1-shot (%) | 5-shot (%) |
| MAML [10] | $51.61_{\pm0.20}$ | $65.76_{\pm0.27}$ | $40.51_{\pm0.08}$ | $53.09_{\pm0.16}$ | $33.57_{\pm0.14}$ | $44.56_{\pm0.21}$ |
| ANIL [38] | $52.82_{\pm0.29}$ | $66.52_{\pm0.28}$ | $41.12_{\pm0.15}$ | $55.82_{\pm0.21}$ | $34.77_{\pm0.31}$ | $46.55_{\pm0.29}$ |
| BOIL [35] | $53.23_{\pm0.41}$ | $69.37_{\pm0.23}$ | $44.20_{\pm0.15}$ | $60.92_{\pm0.11}$ | $36.12_{\pm0.29}$ | $50.64_{\pm0.22}$ |
| SparseMAML+ [56] | $53.91_{\pm0.67}$ | $69.92_{\pm0.21}$ | $43.43_{\pm1.04}$ | $62.02_{\pm0.78}$ | $37.14_{\pm0.77}$ | $53.18_{\pm0.44}$ |
| GAP [19] | $58.56_{\pm0.93}$ | $\mathbf{72.82_{\pm0.77}}$ | $44.74_{\pm0.75}$ | $\mathbf{64.88_{\pm0.72}}$ | $38.44_{\pm0.77}$ | $55.04_{\pm0.77}$ |
| MetaNCoV | $\mathbf{61.50_{\pm1.49}}$ | $\mathbf{73.10_{\pm0.74}}$ | $\mathbf{47.84_{\pm1.49}}$ | $\mathbf{65.27_{\pm0.73}}$ | $\mathbf{41.66_{\pm1.48}}$ | $\mathbf{57.19_{\pm0.75}}$ |

the meta-training and meta-testing phases. By shifting the task domain, this test aims to assess the overfitting of the learned prior to a specific domain. Our test setup follows from [35], where the prior model is meta-trained on the miniImageNet [55] dataset, and meta-tested on tieredImageNet [42], Cars [23], and CUB [57] datasets. Our MetaNCoV is implemented with MetaSGD [29] in this test. As shown in Table 5, our method consistently outperforms popular meta-learning approaches in such a setup, especially in the 1-shot case. This not only confirms the cross-domain generalization of MetaNCoV, but again justifies the importance of expressive prior when data are exceedingly limited.

## 5 Conclusions and outlook

An informative prior plays a crucial role in training a large-scale model with limited small-scale data. This work introduced a novel NCoV model for learning an expressive task-invariant prior. By transforming a known pdf of a continuous random vector, the NCoV model enables a large family of target pdfs. As a flexible plug-in prior model, our MetaNCoV method offers enhanced prior expressiveness compared to existing meta-learning methods that rely on preselected prior pdfs. Numerical studies validate our theoretical analysis, and highlight the superior performance of the proposed method, especially when datasets are scarce. Our future research agenda includes i) investigation of more generic universal approximation theorems; ii) bilevel convergence analysis for the MetaNCoV method; and, iii) implementation of MetaNCoV with alternative transformations, backbone models, and meta-learning methods.

## Acknowledgments

This work was supported by NSF grants 2220292, 2312547, 2212318, and 2126052.

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

# A    Proof of Theorem 3.1

**Theorem A.1** (Multivariate PIT, restated). *Let $P_{\mathbf{Z}} : \mathbb{R}^d \mapsto [0, 1]$ be the cdf of continuous random vector $\mathbf{Z} := [Z_1, \ldots, Z_d]^\top$ with $\{Z_i\}_{i=1}^d$ mutually independent. For any differentiable a.e. cdf $Q : \mathbb{R}^d \mapsto [0, 1]$, there exists a non-decreasing function $f^* : \mathbb{R}^d \mapsto \mathbb{R}^d$ that the random vector $\mathbf{Z}' := f^*(\mathbf{Z})$ has cdf*

$$P_{\mathbf{Z}'} = Q \ (a.e.). \tag{12}$$

*Proof.* We claim that the $i$-th entry of transformation $f^*$ adopts the form

$$f_i^* = g_i \circ h_i, \ i = 1, \ldots, d \tag{13}$$

where $g_i : \mathbb{R}^i \mapsto \mathbb{R}$ will be specified soon, and $h_i(\mathbf{Z}) := [P_{Z_1}(Z_1), \ldots, P_{Z_i}(Z_i)]^\top$. The proof follows from the mathematical induction on $d$.

First consider the base case $d = 1$. By univariate probability transformation, $f^* = Q^{-1} \circ P_{\mathbf{Z}}$ directly verifies Theorem 3.1 and the claim (13), where $Q^{-1}(u) := \inf\{v \mid Q(v) \geq u\}$, $u \in [0, 1]$. What remains is the proof for the monotonicity of $f^*$. Since $Q$ and $P_{\mathbf{Z}}$ are cdfs, they are non-decreasing by definition. Using the monotonicity of $Q$, it holds that $\{v \mid Q(v) \geq u_1\} \subseteq \{v \mid Q(v) \geq u_2\}$, $\forall u_1 \geq u_2$. From the definition of $Q^{-1}$, we have $Q^{-1}(u_1) \geq Q^{-1}(u_2)$, $\forall u_1 \geq u_2$, meaning that $Q^{-1}$ is also non-decreasing. As a result, the composition $f^* = Q^{-1} \circ P_{\mathbf{Z}}$ is non-decreasing.

Subsequently, assuming Theorem 3.1 and the validity of claim (13) for $d = 1, \ldots, d_0$, we establish them for $d = d_0 + 1$. This induction-based argument gives the proof of Theorem 3.1. For notational compactness, define random variable $U_i := P_{Z_i}(Z_i)$, $i = 1, \ldots, d_0 + 1$, and likewise random vector $\mathbf{U}_{1:i} := [U_1, \ldots, U_i] = h_i(\mathbf{Z})$. The univariate PIT indicates that each $U_i \sim \mathcal{U}[0, 1]$. Besides, since $\{Z_i\}_{i=1}^{d_0+1}$ are mutually independent, it follows that $\{U_i\}_{i=1}^{d_0+1}$ are also mutually independent.

Let $\tilde{f} : \mathbb{R}^{d_0} \mapsto \mathbb{R}$ denote the transformation provided by the inductive hypothesis for case $d = d_0$. The first $d_0$ entries of the desired $f^*$ (for case $d = d_0 + 1$) can be defined as $f_{1:d_0}^*(\mathbf{Z}) := \tilde{f}(\mathbf{Z}_{1:d_0})$, $\mathbf{Z} \in \mathbb{R}^{d_0+1}$. Thus, the inductive hypothesis suggest the joint cdf for $\mathbf{Z}'_{1:d_0} = f_{1:d_0}^*(\mathbf{Z}) \in \mathbb{R}^{d_0}$ is $P_{\mathbf{Z}'_{1:d_0}}(\boldsymbol{\xi}) = Q([\boldsymbol{\xi}^\top, +\infty]^\top)$, $\boldsymbol{\xi} \in \mathbb{R}^{d_0}$ (notice that $Q(\cdot)$ is a function defined on $\mathbb{R}^{d_0+1}$ for $d = d_0 + 1$, while $Q([\cdot, +\infty]^\top)$ is on $\mathbb{R}^{d_0}$).

Next, it will be shown that (12) can be obtained upon defining $f_{d_0+1}^* = g_{d_0+1} \circ h_{d_0+1}$ with

$$\begin{aligned}
g_{d_0+1}(\mathbf{U}_{1:d_0+1}) &:= Q_{d_0+1|1:d_0}^{-1}(U_{d_0+1} \mid [g_1(U_1), \ldots, g_{d_0}(\mathbf{U}_{1:d_0})]^\top) \\
&:= \inf\left\{v \mid Q_{d_0+1|1:d_0}(v \mid [g_1(U_1), \ldots, g_{d_0}(\mathbf{U}_{1:d_0})]^\top) \geq U_{d_0+1}\right\} \\
&\stackrel{(a)}{=} \min\left\{v \mid Q_{d_0+1|1:d_0}(v \mid [g_1(U_1), \ldots, g_{d_0}(\mathbf{U}_{1:d_0})]^\top) \geq U_{d_0+1}\right\}
\end{aligned} \tag{14}$$

where conditional cdf $Q_{d_0+1|1:d_0}(v_{d_0+1}|\mathbf{v}_{1:d_0}) := \Pr(V'_{d_0+1} \leq v_{d_0+1} \mid \mathbf{V}_{1:d_0} \preceq \mathbf{v}_{1:d_0})$ for $(d_0+1)$-dimensional random vector $\mathbf{V}$ obeying cdf $Q$, and $(a)$ is because the conditional cdf is non-decreasing and right continuous so the infimum can be attained. First notice that the transformed random vector

$$\mathbf{Z}' = f^*(\mathbf{Z}) = [g_1 \circ h_1(\mathbf{Z}), \ldots, g_{d_0+1} \circ h_{d_0+1}(\mathbf{Z})]^\top = [g_1(U_1), \ldots, g_{d_0+1}(\mathbf{U}_{1:d_0+1})]^\top \tag{15}$$

has cdf

$$P_{\mathbf{Z}'}(\mathbf{v}) = \int_{-\infty}^{\mathbf{v}_{1:d_0}} p_{\mathbf{Z}'_{1:d_0}}(\boldsymbol{\xi}) P_{Z'_{d_0+1}|\mathbf{Z}'_{1:d_0}}(v_{d_0+1} \mid \boldsymbol{\xi}) d\boldsymbol{\xi}, \ \mathbf{v} \in \mathbb{R}^{d_0+1} \tag{16}$$

where the equality is due to $P_{XY}(x, y) = \int_{-\infty}^x p_X(\xi) P_{Y|X}(y|\xi) d\xi$ for random variables $X, Y$.

On one hand, it holds that

$$P_{Z'_{d_0+1}|\mathbf{Z}'_{1:d_0}}(v_{d_0+1} \mid \boldsymbol{\xi}) = \Pr\left(Z'_{d_0+1} \leq v_{d_0+1} \mid \mathbf{Z}'_{1:d_0} = \boldsymbol{\xi}\right)$$

$$\overset{(a)}{=} \Pr(g_{d_0+1}(\mathbf{U}_{1:d_0+1}) \leq v_{d_0+1} \mid g_1(U_1) = \xi_1, \ldots, g_{d_0}(\mathbf{U}_{1:d_0}) = \xi_{d_0})$$

$$\overset{(b)}{=} \Pr\left(Q^{-1}_{d_0+1|1:d_0}(U_{d_0+1}|\boldsymbol{\xi}) \leq v_{d_0+1} \mid g_1(U_1) = \xi_1, \ldots, g_{d_0}(\mathbf{U}_{1:d_0}) = \xi_{d_0}\right)$$

$$\overset{(c)}{=} \Pr\left(Q^{-1}_{d_0+1|1:d_0}(U_{d_0+1}|\boldsymbol{\xi}) \leq v_{d_0+1}\right)$$

$$= \Pr\left(U_{d_0+1} \leq Q_{d_0+1|1:d_0}(v_{d_0+1}|\boldsymbol{\xi})\right)$$

$$\overset{(d)}{=} Q_{d_0+1|1:d_0}(v_{d_0+1}|\boldsymbol{\xi}) \tag{17}$$

where $(a)$ is from (15), $(b)$ uses (14), $(c)$ follows from the mutual independency of $\{U_i\}_{i=1}^{d_0+1}$, and $(d)$ is because $U_{d_0+1} \sim \mathcal{U}[0,1]$.

On the other hand, it has already been shown using the inductive hypothesis that, the random vector $\mathbf{Z}'_{1:d_0} = f^*_{1:d_0}(\mathbf{Z})$ has cdf $P_{\mathbf{Z}'_{1:d_0}}(\boldsymbol{\xi}) = Q([\boldsymbol{\xi}^\top, +\infty]^\top)$, $\boldsymbol{\xi} \in \mathbb{R}^{d_0}$. Since $Q$ is differentiable a.e., we have the corresponding pdf $p_{\mathbf{Z}'_{1:d_0}}(\boldsymbol{\xi}) = \int_{\mathbb{R}} q([\boldsymbol{\xi}^\top, \eta]^\top) d\eta := q_{1:d_0}(\boldsymbol{\xi})$ a.e.. As a result, it follows from (16) and (17) that

$$P_{\mathbf{Z}'}(\mathbf{v}) = \int_{-\infty}^{\mathbf{v}_{1:d_0}} q_{1:d_0}(\boldsymbol{\xi}) Q_{d_0+1|1:d_0}(v_{d_0+1}|\boldsymbol{\xi}) d\boldsymbol{\xi} = Q(\mathbf{v}) \text{ (a.e.)} \tag{18}$$

where we use $P_{XY}(x,y) = \int_{-\infty}^{x} p_X(\xi) P_{Y|X}(y|\xi) d\xi$ again. It should be noted that the only type of discontinuities for a monotone function is the jump discontinuity and there are at most countably many of them. Consequently, $P_{\mathbf{Z}'}$ may fail to match $Q$ only on a set of measure zero.

Finally, we will prove the monotonicity of this constructed $f^*$ by showing that $J_{f^*} \succeq \mathbf{0}_{(d_0+1)\times(d_0+1)}$. First notice from (14) that $g_{d_0+1}(\mathbf{U}_{1:d_0+1})$ is a conditional cdf non-decreasing w.r.t. $U_{d_0+1}$. By the definition of $h_{d_0+1}$, we have $\frac{\partial [h_{d_0+1}(\mathbf{Z})]_{d_0+1}}{\partial Z_{d_0+1}} = P'_{Z_{d_0+1}}(Z_{d_0+1}) \geq 0$ because $P_{Z_{d_0+1}}$ is a cdf. As a result, applying the chain rule leads to $\frac{\partial f^*_{d_0+1}(\mathbf{Z})}{\partial Z_{d_0+1}} = \frac{\partial g_{d_0+1}(h_{d_0+1})}{\partial [h_{d_0+1}]_{d_0+1}} \frac{\partial [h_{d_0+1}(\mathbf{Z})]_{d_0+1}}{\partial Z_{d_0+1}} = \frac{\partial g_{d_0+1}(\mathbf{U}_{1:d_0+1})}{\partial U_{d_0+1}} \frac{\partial [h_{d_0+1}(\mathbf{Z})]_{d_0+1}}{\partial Z_{d_0+1}} \geq 0$. Additionally, the inductive hypothesis implies $\tilde{f}$ is non-decreasing on $\mathbb{R}^d$; that is $J_{\tilde{f}} \succeq \mathbf{0}_{d_0 \times d_0}$. To the end, $f^*(\mathbf{Z}) := [\tilde{f}(\mathbf{Z}_{1:d_0})^\top, f^*_{d_0+1}(\mathbf{Z})]^\top$ has a block triangular Jacobian

$$J_{f^*}(\mathbf{Z}) = \begin{bmatrix} J_{\tilde{f}}(\mathbf{Z}) & \frac{\partial f^*_{d_0+1}(\mathbf{Z})}{\partial \mathbf{Z}_{1:d_0}} \\ \mathbf{0}_{d_0}^\top & \frac{\partial f^*_{d_0+1}(\mathbf{Z})}{\partial Z_{d_0+1}} \end{bmatrix}.$$

It has been shown that diagonal blocks $J_{\tilde{f}}(\mathbf{Z}) \succeq \mathbf{0}_{d_0 \times d_0}$ and $\frac{\partial f^*_{d_0+1}(\mathbf{Z})}{\partial Z_{d_0+1}} \geq 0$, Thus, it follows that $J_{f^*} \succeq \mathbf{0}_{(d_0+1)\times(d_0+1)}$, which completes the proof. $\square$

## B  Proof of Theorem 3.5

To aid the proof of Theorem 3.5, the following lemma offers an alternative expression for (4).

**Lemma B.1.** *Consider the notational conventions of Theorem 3.1. Let $E \subseteq \mathbb{R}^d$ be the set on which $f^*$ is injective and right-continuous. Then, it holds a.e. that*

$$P_{\mathbf{Z}}(\mathbf{z}) = Q \circ f^*(\mathbf{z}), \ \forall \mathbf{z} \in E. \tag{19}$$

*Proof.* With (4) in effect, it holds a.e. that

$$P_{\mathbf{Z}}(\mathbf{z}) = \Pr(\mathbf{Z} \preceq \mathbf{z}) \overset{(a)}{=} \Pr(f^*(\mathbf{Z}) \preceq f^*(\mathbf{z})) = P_{\mathbf{Z}'}(f^*(\mathbf{z})) = Q(f^*(\mathbf{z})) = (Q \circ f^*)(\mathbf{z}) \tag{20}$$

where $(a)$ is because $f^*$ is injective thus increasing and right-continuous on $E$. This proves (19). $\square$

The next lemma suggests the cdf of tail-convergent random vector can be approximated by truncating its pdf on a sufficiently large compact set.

**Lemma B.2.** *For any tail-convergent random vector with cdf $P : \mathbb{R}^d \mapsto [0,1]$, and $\forall \epsilon > 0$, there exists a cdf $\tilde{P} : \mathbb{R}^d \mapsto [0,1]$ for which the pdf $\tilde{p}$ vanishes outside a compact set $E \subset \mathbb{R}^d$, and*

$$|P(\mathbf{v}) - \tilde{P}(\mathbf{v})| < \epsilon, \ \forall \mathbf{v} \in \mathbb{R}^d. \tag{21}$$

*Proof.* For a tail-convergent random vector, Definition 3.4 suggests that for $\forall \epsilon > 0$, there exists a bounded $E' \subset \mathbb{R}^d$ such that $\int_{\mathbb{R}^d \setminus E'} p < \epsilon/2$. Taking $E := \mathrm{cl}(E')$ to be the closure, it follows from the definition of closure that $E$ is compact and

$$\int_{\mathbb{R}^d \setminus E} p \overset{(a)}{\leq} \int_{\mathbb{R}^d \setminus E'} p < \epsilon/2. \tag{22}$$

where $(a)$ is due to $E' \subseteq E$. Now define

$$\tilde{p} := \begin{cases} p/(1 - \int_{\mathbb{R}^d \setminus E} p), & \text{on } E \\ 0, & \text{otherwise} \end{cases}. \tag{23}$$

Notice that $\int_{\mathbb{R}^d} \tilde{p} = \int_E \tilde{p} = \int_E p/(1 - \int_{\mathbb{R}^d \setminus E} p) = 1$, which verifies $\tilde{p}$ is a valid pdf. Thus, the induced cdf is

$$\tilde{P}(\mathbf{v}) = \int_{\{\boldsymbol{\xi}|\boldsymbol{\xi} \preceq \mathbf{v}\}} \tilde{p}(\boldsymbol{\xi}) d\boldsymbol{\xi}. \tag{24}$$

It then follows for $\forall \mathbf{v} \in \mathbb{R}^d$ that

$$
\begin{aligned}
|P(\mathbf{v}) - \tilde{P}(\mathbf{v})| &= \left| \int_{\{\boldsymbol{\xi}|\boldsymbol{\xi} \preceq \mathbf{v}\}} p - \tilde{p}(\boldsymbol{\xi}) d\boldsymbol{\xi} \right| \\
&= \left| \int_{\{\boldsymbol{\xi}|\boldsymbol{\xi} \preceq \mathbf{v}\} \cap E} p(\boldsymbol{\xi}) - \tilde{p}(\boldsymbol{\xi}) d\boldsymbol{\xi} \right| + \left| \int_{\{\boldsymbol{\xi}|\boldsymbol{\xi} \preceq \mathbf{v}\} \setminus E} p(\boldsymbol{\xi}) - \tilde{p}(\boldsymbol{\xi}) d\boldsymbol{\xi} \right| \\
&\overset{(a)}{=} \frac{\int_{\mathbb{R}^d \setminus E} p(\boldsymbol{\xi}) d\boldsymbol{\xi}}{1 - \int_{\mathbb{R}^d \setminus E} p(\boldsymbol{\xi}) d\boldsymbol{\xi}} \int_{\{\boldsymbol{\xi}|\boldsymbol{\xi} \preceq \mathbf{v}\} \cap E} p(\boldsymbol{\xi}) d\boldsymbol{\xi} + \int_{\{\boldsymbol{\xi}|\boldsymbol{\xi} \preceq \mathbf{v}\} \setminus E} p(\boldsymbol{\xi}) d\boldsymbol{\xi} \\
&\leq \frac{\int_{\mathbb{R}^d \setminus E} p(\boldsymbol{\xi}) d\boldsymbol{\xi}}{1 - \int_{\mathbb{R}^d \setminus E} p(\boldsymbol{\xi}) d\boldsymbol{\xi}} \int_E p(\boldsymbol{\xi}) d\boldsymbol{\xi} + \int_{\mathbb{R}^d \setminus E} p(\boldsymbol{\xi}) d\boldsymbol{\xi} \\
&= \int_{\mathbb{R}^d \setminus E} p(\boldsymbol{\xi}) d\boldsymbol{\xi} + \int_{\mathbb{R}^d \setminus E} p(\boldsymbol{\xi}) d\boldsymbol{\xi} \tag{25} \\
&\overset{(b)}{\leq} \epsilon/2 + \epsilon/2 = \epsilon. \tag{26}
\end{aligned}
$$

where $(a)$ uses (23), and $(b)$ is from (22). $\qquad\square$

Next, the classic universal approximation theorem will be generalized to suit for the case of NCoVs.

**Definition B.3** ([5]). A function $\sigma : \mathbb{R} \mapsto \mathbb{R}$ is said to be **sigmoidal** if

$$\sigma(t) \to \begin{cases} 1, & \text{as } t \to +\infty \\ 0, & \text{as } t \to -\infty \end{cases}. \tag{27}$$

**Definition B.4** ([5], generalized). Let $E$ be a compact set with positive Borel measure. A function $\sigma : \mathbb{R} \mapsto \mathbb{R}$ is said to be **discriminatory on** $E$ if for a finite signed regular Borel measure $\mu$, it holds that

$$\int_E \sigma(\mathbf{b}^\top \mathbf{z} + c) d\mu(\mathbf{z}) = 0 \tag{28}$$

for all $\mathbf{b} \in \mathbb{R}^d$ and $c \in \mathbb{R}$ implies $\mu = 0$.

**Theorem B.5** ([5, Theorem 1]). *Let $\sigma$ be a bounded measurable sigmoidal function. Then finite sum of the form*

$$G(\mathbf{z}) = \sum_{j=1}^N a_j \sigma(\mathbf{b}_j^\top \mathbf{z} + c_j) \tag{29}$$

*is dense in $C([0,1]^d)$.*

**Corollary B.6.** *Let $\sigma$ be a bounded measurable sigmoidal function, and $E \subset \mathbb{R}^d$ a locally compact set of finite Borel measure. Then finite sum of the form*

$$G(\mathbf{z}) = \sum_{j=1}^{N} a_j \sigma(\mathbf{b}_j^\top \mathbf{z} + c_j) \tag{30}$$

*is dense in $C(E)$.*

*Proof.* When $\mu(E) = 0$, one can take $E_0 = E$ and the Corollary holds trivially. Next we consider the case $\mu(E) > 0$.

The original proof of Theorem B.5 relies on Lebesgue Bounded Convergence Theorem, Hahn-Banach theorem, and Riesz Representation Theorem. All these four theorems hold for a locally compact set $E$ with finite Borel measure.

The proof of Corollary B.6 follows by i) generalizing the definition of **discriminatory** in [5] to definition B.4, and ii) replacing $[0, 1]^d$ in the proof of Theorem B.5 with $E$. $\qquad\square$

Building upon Lemma B.1, Lemma B.2, and Corollary B.6, the proof of Theorem 3.5 is provided as follows.

**Theorem B.7** (Universal approximation via Sylvester NCoVs, restated). *Let $P_\mathbf{Z}$ denote the cdf of tail-convergent continuous random vector $\mathbf{Z} \in \mathbb{R}^d$ with mutually independent entries, and $Q$ a Lipschitz cdf of a tail-convergent random vector. For any $\epsilon > 0$, there exists cdfs $\tilde{P}, \tilde{Q}$ for which the pdfs $\tilde{p}, \tilde{q}$ vanishes outside compact sets $E_p, E_q$, and*

$$|P_\mathbf{Z}(\mathbf{v}) - \tilde{P}(\mathbf{v})| < \epsilon, \ |Q_\mathbf{Z}(\mathbf{v}) - \tilde{Q}(\mathbf{v})| < \epsilon, \ \forall \mathbf{v} \in \mathbb{R}^d. \tag{31}$$

*Moreover, let $E \subseteq E_p$ be any set on which the transform $f^*$ matching $\tilde{P}_\mathbf{Z}$ to $\tilde{Q}$ (cf. Theorem 3.1) is injective and right-continuous. There exists a Sylvester NCoV $f$ and a zero-measure set $E_0$, such that*

$$|f(\mathbf{Z}) - f^*(\mathbf{Z})| < \epsilon, \ \forall \mathbf{Z} \in E_p \setminus E_0, \tag{32a}$$
$$|P_\mathbf{Z}(\mathbf{z}) - Q \circ f(\mathbf{z})| < \epsilon, \ \forall \mathbf{z} \in E \setminus E_0. \tag{32b}$$

*Proof.* Since $\mathbf{Z}$ is tail-convergent, Lemma B.2 suggests that there exists a cdf $\tilde{P}_\mathbf{Z}$ for which the pdf $\tilde{p}_\mathbf{Z}$ vanishes outside a compact set $E_P \subset \mathbb{R}^d$, and $|P_\mathbf{Z}(\mathbf{z}) - \tilde{P}_\mathbf{Z}(\mathbf{z})| < \epsilon/4$, $\forall \mathbf{z} \in \mathbb{R}^d$. Similarly, there is also a cdf $\tilde{Q}$ for which $\tilde{q}$ is supported on a compact set $E_Q$, and $|Q(\mathbf{z}) - \tilde{Q}(\mathbf{z})| < \epsilon/4$, $\forall \mathbf{z} \in \mathbb{R}^d$.

Moreover, let $L_Q$ be the Lipschitz constant of $Q$. Then, (22), (23) and (24) indicates $\tilde{Q}$ is also Lipschitz with constant

$$L_Q / \left(1 - \int_{\mathbb{R}^d \setminus E_Q} q\right) < \frac{1}{L_Q - \epsilon/8}. \tag{33}$$

Using Rademacher theorem, we have $\tilde{Q}$ differentiable a.e.. Then, let $f^* : \mathbb{R}^d \mapsto \mathbb{R}^d$ denote the optimal transform by Theorem 3.1, which matches $\tilde{P}$ to $\tilde{Q}$. Lemma B.1 suggests that

$$\tilde{P}_\mathbf{Z}(\mathbf{z}) = \tilde{Q} \circ f^*(\mathbf{z}), \ \forall \mathbf{z} \in E. \tag{34}$$

Let $\tilde{\mathbf{Z}}$ and $\tilde{\mathbf{Z}}' = f^*(\tilde{\mathbf{Z}})$ be random vectors obeying cdfs $\tilde{P}_\mathbf{Z}$ and $\tilde{Q}$. Since $\tilde{p}_\mathbf{Z}$ is supported on $E_P$, (5) implies that $f^*$ can have arbitrary value outside $E_P$, which will not change $\tilde{p}_{\mathbf{Z}'}$.

We assert that $f^*$ is bounded on $E_P$. Otherwise, for any $B > 0$, there must be some $\mathbf{z}_0 \in E_P$ such that $\|f^*(\mathbf{z}_0)\| > B$, and using (5) that

$$\tilde{q}(f^*(\mathbf{z}_0)) = \tilde{p}_{\mathbf{Z}'}(f^*(\mathbf{z}_0)) = \int \tilde{p}_\mathbf{Z}(\mathbf{z})\delta[f^*(\mathbf{z}_0) - f^*(\mathbf{z})]d\mathbf{z}$$

$$\geq \int \tilde{p}_\mathbf{Z}(\mathbf{z})\delta[\mathbf{z}_0 - \mathbf{z}]d\mathbf{z}$$

$$= \tilde{p}_\mathbf{Z}(\mathbf{z}_0) > 0 \tag{35}$$

where the inequality is because $f^*$ is non-decreasing. Since $B$ can be arbitrarily large, this contradicts with the fact that $\text{supp}(\tilde{q}) = E_Q$ is compact (cf. Lemma B.2).

Moreover, the monotonicity of $f^*$ indicates the only possible discontinuities of it must be jump discontinuities, and there are at most countably many of them. Let $E_0 \subseteq E_p$ be the set where $f^*$ is discontinuous. From the countability of $E_0$ we have $\mu(E_0) = 0$. Thus, $f_i^*(\mathbf{z}) - z_i$ is bounded and continuous on $E_p \setminus E_0$, where $f_i^*$ and $z_i$ are the $i$-th entries of $f^*$ and $\mathbf{z}$.

Then, applying Corollary B.6 implies that there exists a $G_i(\mathbf{z})$ of form (30) such that

$$|G_i(\mathbf{z}) - [f_i^*(\mathbf{z}) - z_i]| < \frac{\epsilon(1 - \epsilon/8)}{2L_Q\sqrt{d}}, \ \forall \mathbf{z} \in E_p \setminus E_0. \tag{36}$$

Now, define $f_i(\mathbf{z}) = G_i(\mathbf{z}) + z_i$, $i = 1, \ldots, d$ on $\mathbb{R}^d$. One can easily verify from (6) that such a definition renders $f$ a Sylvester NCoV, and (9a) holds. Moreover, it follows for $\forall \mathbf{z} \in E_p \setminus E_0$ that

$$
\begin{aligned}
|P_{\mathbf{Z}}(\mathbf{z}) - Q \circ f(\mathbf{z})| &\leq |\tilde{P}_{\mathbf{Z}}(\mathbf{z}) - \tilde{Q} \circ f(\mathbf{z})| + |P_{\mathbf{Z}}(\mathbf{z}) - \tilde{P}_{\mathbf{Z}}(\mathbf{z})| + |Q \circ f(\mathbf{z}) - \tilde{Q} \circ f(\mathbf{z})| \\
&< |\tilde{P}_{\mathbf{Z}}(\mathbf{z}) - \tilde{Q} \circ f(\mathbf{z})| + \epsilon/4 + \epsilon/4 \\
&\overset{(a)}{=} |\tilde{Q} \circ f^*(\mathbf{z}) - \tilde{Q} \circ f(\mathbf{z})| + \epsilon/2 \\
&\overset{(b)}{\leq} \frac{L_Q}{1 - \epsilon/8}\|f^*(\mathbf{z}) - f(\mathbf{z})\|_2 + \epsilon/2 \\
&\overset{(c)}{\leq} \epsilon/2 + \epsilon/2 = \epsilon
\end{aligned}
\tag{37}
$$

where $(a)$ follows from (34), $(b)$ is due to (33), and $(c)$ uses (36). The proof is thus completed. $\square$

## C  Detailed setups of numerical tests

Our codes are run on a server equipped with an Intel Core i7-12700 CPU, and an NVIDIA RTX A5000 GPU. The experimental setups adopted in this paper will be next elaborated.

### C.1  Toy tests

The numerical tests for the 2D toy examples demonstrated in Figure 1 are carried over by training a Sylvester NCoV with a width $m = 50$. We used SGD optimizer with learning rate of $10^{-3}$ and momentum of $0.9$. The ground truth samples used to train the model were generated by transforming 2D Gaussian random vectors $\mathbf{Z} \sim \mathcal{N}(\mathbf{0}_{2\times 1}, \mathbf{I}_{2\times 2})$ through a non-injective ground truth transformation $f^*(\mathbf{Z}) = \mathbf{A}\,\sigma(\mathbf{B}\sin(\mathbf{Z}) + \mathbf{c})$, where $\sin(\cdot)$ function is applied element-wise to each dimension of vector $\mathbf{Z}$ separately. A set of i.i.d. samples $\{\mathbf{z}_i, f^*(\mathbf{z}_i)\}_{i=1}^{10^5}$ was randomly generated and used to train the Sylvester NCoV model. The results presented in Figure 1 were obtained using three different settings of ground truth $f^*(\cdot)$. In each of these settings, the elements of the underlying matrices $\mathbf{A}, \mathbf{B}$, and the vector $\mathbf{c}$ were generated from Gaussian distributions with zero mean and unit variance. The result for 1D case presented in Figure 2 was obtained using a smaller Sylvester NCoV with width $m = 3$, trained using SGD with learning rate of $10^{-2}$ and momentum of $0.6$, and the histogram of the generated samples was normalized to represent a pdf. To generate training samples, we employed the probability integral transform (PIT). Specifically, we first draw ground truth $Z'$ from a mixture of Gaussians, denoted as $Z' \sim q(z')$, where $q := \sum_{k=1}^2 \frac{1}{2}\mathcal{N}(\mu_k, \sigma_k^2)$. Here, $\mu_1 = -10$ and $\mu_2 = 10$, with $\sigma_1^2 = \sigma_2^2 = 1$. Then, we rely on the inverse transformation $f^{*-1}(Z')$ to find its paired $Z$, where $f^*(Z) := (Q^{-1} \circ P_Z)(Z)$ is the ground truth transformation obtained via PIT. Having find this mapping, we draw a set of i.i.d. samples $\{z_i, f^*(z_i)\}_{i=1}^{10^5}$ to train Sylvester NCoV model in 1D.

### C.2  Few-shot classifications

A brief description of the three benchmark datasets used in our experiments are provided next.

**MiniImageNet** [55] contains $60,000$ images sampled from the full ImageNet (ILSVRC-12) dataset, which are divided into 100 classes, each with 600 instances. All images are cropped and resized into

Table 6: Hyperparameter setups.

| Hyperparameter | Notation | Value |
|---|---|---|
| Task-level iterations | $K$ | 5 |
| Task-level learning rate (ConvNet-4) | $\alpha$ | $10^{-2}$ |
| Task-level learning rate (WRN-28-10) | $\alpha$ | 2 |
| Meta-level iterations | $R$ | $60,000$ |
| Meta-level learning rate | $\beta$ | $10^{-3}$ |
| Meta-level SGD batch size | $B$ | 4 |

$84 \times 84$ pixels. In the experiments, we adopt the dataset split suggested by [41], where 64, 16 and 20 disjoint classes can be accessed during the training, validation, and testing phases of meta-learning.

**TieredImageNet** [42] is a larger subset of the ImageNet dataset, composed of $779,165$ images from 608 classes. Likewise, all the images are preprocessed to have size $84 \times 84$. Instead of using a random split, classes are partitioned into 34 categories according to the hierarchy of ImageNet dataset. Each category contains 10 to 30 classes. These categories are further grouped into 3 different sets: 20 for training, 6 for validation, and 8 for testing.

**CUB-200-2011** [57] is an extended version of the Caltech-UCSD Birds(CUB)-200 dataset, which consists of $11,788$ fine-grained images from 200 bird species. The dataset split follows from [4], dividing the species into 100 training, 50 validation, and 50 testing classes. Similar to the preceding two datasets, the images are also resized to $84 \times 84$.

The hyperparameters used for the few-shot classification experiments are the same as those in MAML [10], which are listed in Table 6. To enhance the statbility of the training process, we use SGD with Nesterov momentum instead of Adam as the optimizer for (10a). The width $m$ of Sylvester NCoV is determined through a grid search using the validation tasks. For miniImageNet dataset with a 4-block CNN model, $m = 10$ in the 1-shot experiment and $m = 5$ in the 5-shot one. For miniImageNet and tieredImageNet with WRN-28-10 embeddings, $m$ is fixed to be 10 under both center and multi-view crops. For the CUB dataset, we use $m = 5$ in all the tests.

## D Additional experiments

### D.1 NCoVs in 1D

Here, we demonstrate the efficacy of Sylvester NCoVs in approximating mixture of Gaussians in one-dimensional (1D) scenario. The primary objective is to transform 1D Gaussian random variables, denoted as $Z \sim \mathcal{N}(0, 1)$, into a mixture of Gaussians using a trained Sylvester NCoV. As depicted in

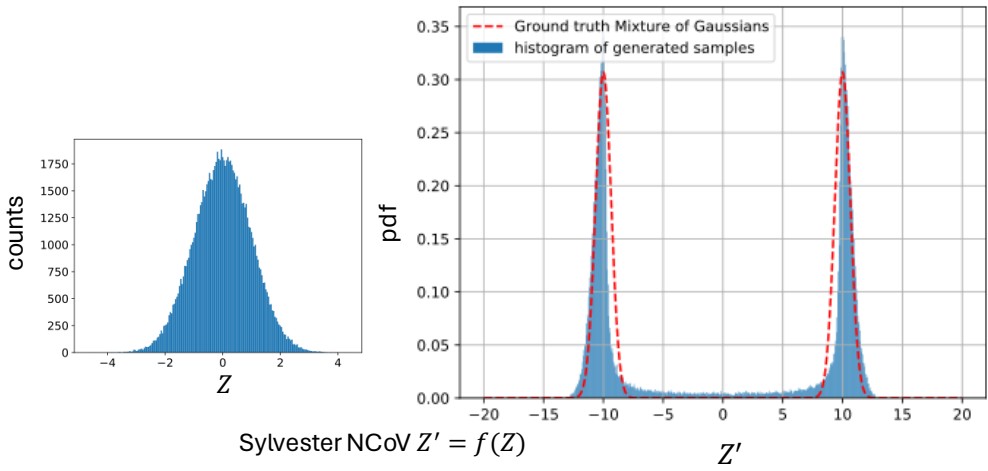

Figure 2: Transforming 1D Gaussian random variable $Z \sim \mathcal{N}(0, 1)$ to a mixture of Gaussians $Z' = f(Z)$ using Sylvester NCoV $f(\cdot)$.

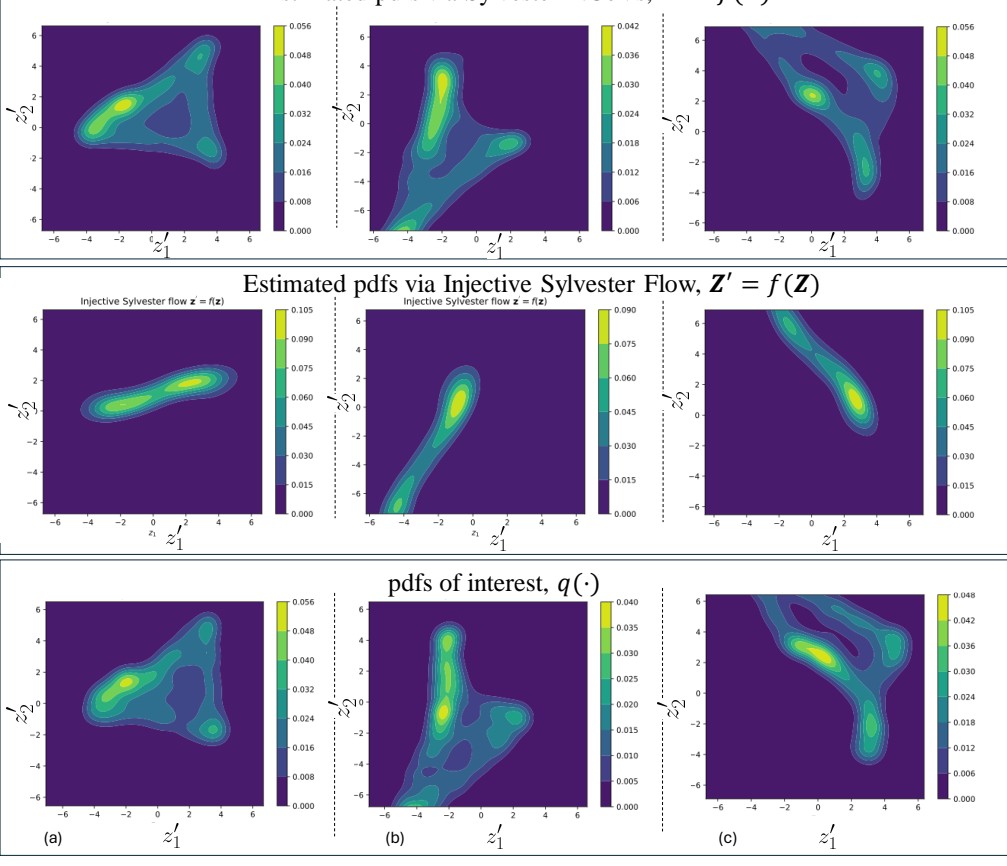

Figure 3: Comparison of NFs and NCoVs in learning 2D toy pdfs.

Figure 2 (left), we illustrate the histogram of the original random variable $Z$ and the estimated pdf of the transformed random variables $Z' = f(Z)$ on the right-hand side. The dashed red curve represents the ground truth mixture of Gaussians that we seek to estimate. This experiment demonstrates the ability of Sylvester NCoVs to effectively transform a basic Gaussian random variable in 1D to the desired mixture of Gaussians $Z' \sim p_{Z'}(z') := \sum_{k=1}^{2} \frac{1}{2} \mathcal{N}(\mu_k, \sigma_k^2)$, where $\mu_1 = -10$, and $\mu_2 = 10$, with $\sigma_1^2 = \sigma_2^2 = 1$.

### D.2 Comparison of NFs and NCoVs using toy data

To demonstrate the expressiveness of NCoVs compared to an injective Sylvester NF, we conducted a 2D toy test showcased in Figure 3. We have employed the QR factorization technique described in [51] to enforce the injectivity constraint for Sylvester NF, which requires $m \leq d$. Here, $m$ is the width of the NF, and $d = 2$ is the dimension of $\mathbf{z}$ for this 2D example. To ensure the injectivity of Sylvester NF, $m$ is thus chosen as 2. For a fair comparison, our NCoV is implemented with the same setup. The three rows in the figure display respectively the learned NCoVs, injective NFs, and the target pdf $q$. It is seen that NCoVs are more expressive than valid NFs when learning complex distributions.

To illustrate the ability of NCoVs in learning pdfs whose support is not full in $\mathbb{R}^d$, we consider a $4 \times 4$ checkerboard pdf. The test has been visualized in Figure 4, where we employ an injective Sylvester NF and a non-injective Sylvester NCoV to learn the target distribution. In our test, we stacked 40 base injective Sylvester NFs, and compared the results with those obtained using a Sylvester NCoV with the same number of parameters and the same training setup. It is observed that NCoV demonstrates remarkable performance in learning the checkerboard pdf compared with that of the injective NF. This observation underscores the expressiveness of NCoVs in effectively learning particular distributions, especially in scenarios involving discrete distributions or those with incomplete support.

Table 7: Performance comparison using a 4-block CNN backbone with different number of channels.

| Method | Channels per block | 5-class miniImageNet | |
| --- | --- | --- | --- |
| | | 1-shot (%) | 5-shot (%) |
| MAML [10] | 32-32-32-32 | $48.70_{\pm 1.84}$ | $63.11_{\pm 0.92}$ |
| MetaSGD [29] | 32-32-32-32 | $50.47_{\pm 1.87}$ | $64.03_{\pm 0.94}$ |
| Meta-LSTM [41] | 64-64-64-64 | $43.44_{\pm 0.77}$ | $60.60_{\pm 0.71}$ |
| MAML [10] | 64-64-64-64 | $49.5_{\pm 1.8}$ | - |
| R2D2 [3] | 64-64-64-64 | $49.5_{\pm 0.2}$ | $65.4_{\pm 0.2}$ |
| MAML + L2F [2] | 64-64-64-64 | $52.10_{\pm 0.50}$ | $69.38_{\pm 0.46}$ |
| Minimax-MAML [58] | 64-64-64-64 | $51.70_{\pm 0.42}$ | $68.41_{\pm 1.28}$ |
| MAML + MetaNCoV (ours) | 64-64-64-64 | $55.86_{\pm 1.49}$ | $68.90_{\pm 0.71}$ |
| MetaSGD + MetaNCoV (ours) | 64-64-64-64 | $57.44_{\pm 1.48}$ | $69.15_{\pm 0.71}$ |
| R2D2 [3] | 96-192-384-512 | $51.8_{\pm 0.2}$ | $68.4_{\pm 0.2}$ |
| MC [37] | 128-128-128-128 | $54.08_{\pm 0.93}$ | $67.99_{\pm 0.73}$ |
| Warp-MAML [12] | 128-128-128-128 | $52.3_{\pm 0.8}$ | $68.4_{\pm 0.6}$ |
| MAML + MetaNCoV (ours) | 128-128-128-128 | $\mathbf{57.74_{\pm 1.47}}$ | $70.72_{\pm 0.70}$ |
| MetaSGD + MetaNCoV (ours) | 128-128-128-128 | $\mathbf{59.10_{\pm 1.52}}$ | $\mathbf{71.48_{\pm 0.68}}$ |

## D.3  Influence of backbones

Next, we show the performance comparison using the 4-block CNN backbone with different number of channels. All the hyperparameters are the same with Appendix C, and the results are gathered in Table 7. One can observe tht reducing the number of channels from 128 to 64 leads to slightly decreased accuracies, yet still superior than all the state-of-the-art competitors thanks to the improved expressive power of NCoV.

## D.4  Complexity and scalability

To demonstrate the scalability of MetaNCoV, this test measures numerically the time and space complexities of MetaNCoV and several popular meta-learning methods during the meta-training phase. The test is conducted on the miniImageNet dataset, and the comparison is conducted using the 4-block CNN with 64-channel and 128-channel setups, respectively. The experimental results are summarized in the Table 8. It is observed that MetaNCoV has a dimension $D$ comparable to MetaSGD, yet notably smaller than MC. One can also see that the increase of $D$ only brings about a marginal growth in both time and space consumption. This is because the key factor affecting the complexity is the Hessian-vector product (HVP) computations when backpropagating the meta-gradient $\nabla_{\boldsymbol{\theta}} \mathcal{L}(\hat{\boldsymbol{\phi}}_t(\boldsymbol{\theta}); \mathcal{D}_t^{\mathrm{val}})$. Since all the methods relied on a $K$-step GD to obtain $\hat{\boldsymbol{\phi}}_t(\boldsymbol{\theta})$ (where MC adopted an additional Kronecker-factorized preconditioner [37]), the complexity for computing this meta-gradient is $\mathcal{O}(Kd)$ in both time and space.

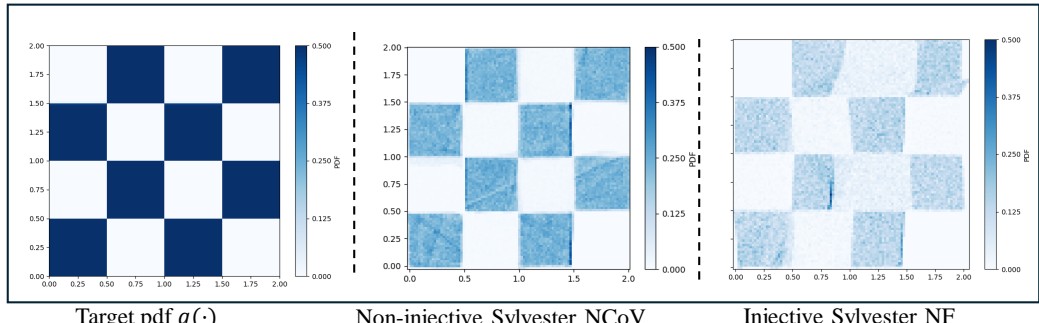

Figure 4: Learning a $4 \times 4$ checkerboard pdf with NFs and NCoVs.

Table 8: Time and Space complexity comparison.

| 64-channel ($d = 121,093$) | MAML | MetaSGD | MC | MAML + MetaNCoV |
|---|---|---|---|---|
| Dimension $D$ of $\boldsymbol{\theta}$ | $121,093$ | $242,186$ | $2,710,896$ | $605,465$ |
| Time (Relative) | $\times 1$ | $\times 1.022$ | $\times 1.189$ | $\times 1.199$ |
| GPU space (MB) | $1,700$ | $1,704$ | $1,726$ | $1,724$ |

| 128-channel ($d = 463,365$) | MAML | MetaSGD | MC | MAML + MetaNCoV |
|---|---|---|---|---|
| Dimension $D$ of $\boldsymbol{\theta}$ | $463,365$ | $926,730$ | $10,819,952$ | $2,316,825$ |
| Time (Relative) | $\times 1.806$ | $\times 1.828$ | $\times 2.023$ | $\times 2.028$ |
| GPU space (MB) | $2,982$ | $3,002$ | $3,052$ | $3,040$ |

# E   Additional related works and comparisons

**Normalizing flows (NFs).** Our NCoV is similar to conventional NFs in the sense that they are both distribution models relying on the change-of-variable formula. Nevertheless, NCoV is distinct from NF in four key aspects: expressiveness, tractability, training strategy, and application fields.

i) First, NCoV has been proved capable of modeling an arbitrary pdf, whereas NF is restricted to pdfs with full support.

ii) Second, while NFs aim for closed-form pdf (3), NCoV sacrifices its tractability for augmented expressiveness.

iii) Third, NFs are typically trained in an unsupervised manner by maximizing the likelihood function $p_{f(\mathbf{z})}$ over a dataset sampled from target pdf $q$. In contrast, NCoV requires latent optimization (10) given the intractability of likelihood.

iv) Finally, NFs can be thus utilized in applications including probability estimation and sample generation, while NCoV is tailored specifically to learn a prior over model parameters in the meta-learning context.

**Non-injective transformations.** Moreover, it is worth mentioning that non-injective transformations have been also considered by [34] to enhance the expressiveness. While [34] allows $\mathbf{Z}$ and $\mathbf{Z}'$ to have different dimensions, it does not provides theoretical guarantees of its expressiveness. In contrast, the design of our MetaNCoV is based on Theorem 3.1 that requires $f$ being a mapping from $\mathbb{R}^d$ to $\mathbb{R}^d$. In addition, other works such as [20] also forgo the analytical invertibility, but typically necessitate an efficient approximation of $f^{-1}$.

**Bayesian meta-learning.** Our approach is also related to Bayesian meta-learning [60, 40, 59, 11]. While these approaches aim to quantify the uncertainties in $\boldsymbol{\phi}_t$ by identifying its posterior $p(\boldsymbol{\phi}|\mathbf{y}_t^{\text{trn}}; \mathbf{X}_t^{\text{trn}}, \boldsymbol{\theta}) \propto p(\mathbf{y}_t^{\text{trn}}|\boldsymbol{\phi}_t; \mathbf{X}_t^{\text{trn}})p(\boldsymbol{\phi}_t; \boldsymbol{\theta})$, our MetaNCoV is deterministic by formulating the task-level optimization (10b) as a MAP problem. In addition, these methods also rely on tractable prior and (surrogate) posterior pdfs of prefixed forms such as Gaussian. For example, [60] relies on a predefined conjugate prior over the extracted features (rather than parameters $\boldsymbol{\phi}_t$) to ensure the tractability of the feature posterior. In [40, Section 3.3], the surrogate (variational) posterior is prespecified as a diagonal Gaussian distribution, while the prior is fixed to be the Gaussian-Gamma form. For [59], particle sampling with SVGD is utilized to parameterize and optimize the prior, where RBF kernels (i.e., Gaussian kernels) are selected to interpolate the particles. As also noted in [11, Section 4.2], both the prior and surrogate posterior distributions are predefined as diagonal Gaussian forms. In contrast, our approach forgoes the tractability for enhanced prior expressiveness.

**Optimal transport.** Lastly, the introduced NCoV model is related to the optimal transport problem in statistics [54], which aims to minimize the total cost of transporting one distribution to another. However, the existence of the transport maps $f$ can impose strong assumptions on both the source and target distributions, and is closely connected to the choice of cost function. For instance, the well-known Brenier's theorem suggests that $f$ exists when $\mathbf{Z}$ and $\mathbf{Z}'$ have finite second moments (or, $q$ has a compact support), and $\mathbf{Z}$ assigns no mass to any set of Hausdorff dimension $d - 1$ [53]. In comparison, Theorem 3.1 puts constraints only on the source random vector $\mathbf{Z}$, as we can choose $P_{\mathbf{Z}}$ flexibly to suit our needs. This enables $P_{Z'}$ to match an arbitrary target $Q$.

