# OpenReview forum: "Meta-Learning Universal Priors Using Non-Injective Change of Variables"
_NeurIPS.cc/2024/Conference — NeurIPS 2024 poster_

### Official Review · Reviewer_XQHx · 2024-07-09

**Soundness:** 3
**Presentation:** 3
**Contribution:** 3
**Rating:** 6
**Confidence:** 3

**Summary:**

The paper proposes to learn a more expressive parameter prior compared to predefined ones for a meta-learner. The problem is formulated in a Bayesian model-agnostic meta-learning context and its solution leverages a class of non-invective normalising flows to learn the parameter prior (aka Sylvester flows). Experiments are conducted on mini-ImageNet and CUB datasets, by comparing the proposed approach applied to existing meta-learning algorithms against their base counterparts, showcasing the benefits of an adaptive prior in terms of few-shot classification performance.

**Strengths:**

1. The paper is overall well written and clear **Clarity**
2. The considered problem of learning a prior is relevant in the context of meta-learning **Relevance**
3. While Sylvester normalising flows are not part of the contribution of the paper, its theoretical analysis gives a nice motivation to their use. Moreover, the theoretical part of the paper seems to be correct and sound **Soundness**
4. Experiments show convincingly that the proposed solution outperforms base methods **Significance**
5. Code is available during submission, demonstrating the openness to share it publicly **Openness**

**Weaknesses:**

1. One of the major concern with the proposed solution is in its novelty and therefore its contribution **Novelty**. There are several variants of probabilistic formulations for model-agnostic meta-learning with corresponding learnt prior. How does the proposed solution differ from them and what is therefore the novel contribution? See for instance [1-3].
2. Experiments miss important baselines **Quality**. See for instance [1-3].

**Reference**

[1] Amortised Bayesian Meta-Learning. ICLR 2019

[2] Bayesian Model-Agnostic Meta-Learning. NeurIPS 2018

[3] Probabilistic Model-Agnostic Meta-Learning. NeurIPS 2018

**Questions:**

In addition to addressing the two above-mentioned weaknesses, can you elaborate more on how the model handles uncertainty, given its probabilistic nature?

**Limitations:**

An important limitation of the proposed solution is its computational complexity. A thorough evaluation and discussion about the trade-off between computation and performance should be also provided.

---

> ### Author Rebuttal · Authors · 2024-08-05
>
> Thank you for the questions. The issues raised are addressed one-by-one next.
>
> **Response to weakness 1**
>
> This contribution differs remarkably from probabilistic meta-learning in three key aspects.
> - Deterministic algorithm. Our MetaNCoV is a *deterministic* meta-learning approach where task-level optimization (10b) is formulated as a maximum a posteriori (MAP) problem. This MAP estimator can be efficiently approximated by gradient descent. In comparison, probabilistic meta-learning methods aim to approximate the intractable posterior $p(\boldsymbol{\phi}_t | \mathbf{y}_t^\mathrm{trn}; \mathbf{X}_t^\mathrm{trn}, \boldsymbol{\theta}) \propto p(\mathbf{y}_t^\mathrm{trn} | \boldsymbol{\phi}_t; \mathbf{X}_t^\mathrm{trn}) p(\boldsymbol{\phi}_t; \boldsymbol{\theta})$ using variational inference [1,3] or particle sampling [2].
> - Prior Expressiveness. While probabilistic meta-learning relies on tractable prior and (surrogate) posterior pdfs of prefixed forms, such as Gaussian [1,3], our approach forgoes the tractability for enhanced prior expressiveness. This allows for a data-driven prior of universal forms beyond the Gaussian family.
> - Theoretical Analysis. The theoretical analysis also distinguishes the current paper with probabilistic meta-learning methods [1-3], which are based on empirical designs.
>
> **Response to weakness 2**
>
> We will add these three additional baselines to Table 1 for a more comprehensive comparison.
>
> **Response to questions and limitations**
>
> Our method is *deterministic* rather than probabilistic, which cannot directly handle uncertainty quantification due to the intractability of the prior pdf, as mentioned in Remark 3.3.
>
> The complexity analysis and performance trade-off have been already investigated in our submission, with results respectively reported in Appendices D.4 and D.3. It can be seen that our MetaNCoV has comparable time and space complexity relative to popular optimization-based meta-learning approaches.
>
> [1] S. Ravi, and A. Beatson, "Amortised Bayesian meta-learning," ICLR 2019.
>
> [2] J. Yoon, T. Kim, O. Dia, S. Kim, Y. Bengio, and S. Ahn, "Bayesian model-agnostic meta-learning," NeurIPS 2018.
>
> [3] C. Finn, K. Xu, and S. Levine, "Probabilistic model-agnostic meta-learning," NeurIPS 2018.

---

> ### Comment · Reviewer_XQHx · 2024-08-11
> **Answer**
>
> Thank you for the clarifying answers, which have addressed some of my questions.  I have some additional questions: Could you please elaborate more on the "prefixed forms" for the above-mentioned probabilistic baselines? Specifically, is the Gaussian assumption imposed both at the prior and posterior level? Moreover, what is the rationale for not considering the above-mentioned methods as important baselines for comparison?

---

> ### Author Response · Authors · 2024-08-11
>
> Thank you for your follow-up questions, which are addressed in the following.
>
> **Regarding prefixed priors in probabilistic meta-learning**
>
> In [Section 3.3, 1], the surrogate (variational) posterior is prespecified as a diagonal Gaussian distribution, while the prior is fixed to be the Gaussian-Gamma form. For [2], particle sampling with SVGD is utilized to parameterize and optimize the prior, where RBF kernels (i.e., Gaussian kernels) are selected in Section 5 to interpolate the particles. As also noted in [Section 4.2, 3] and [Algorithm 1, 3], both the prior and surrogate posterior distributions are predefined as diagonal Gaussian forms.
>
> **Regarding the exclusion of probabilistic methods**
>
> The primary rationales for excluding these probabilistic methods is their high computational complexities compared to deterministic ones. In particular, [1] and [3] rely on variational inference, which requires sampling from the surrogate posterior $q(\boldsymbol{\phi}_t; \mathbf{v}_t)$ (with variational parameter $\mathbf{v}_t$) at each optimization step to calculate the expected NLL $\mathbb{E} _{q(\boldsymbol{\phi}_t; \mathbf{v}_t)} -\log p (\mathbf{y}_t^\mathrm{trn} | \boldsymbol{\phi}_t; \mathbf{X}_t^\mathrm{trn})$. In [2], a group of particles is maintained to parameterized the parameter distribution. As a result, these probabilistic methods [1-3] suffer from $M\times$ time- and space-complexity compared to deterministic ones, where $M$ is the number of samples.
>
> However, it is worth noting that our MetaNCoV consistently outperforms these probabilistic methods by a significant margin. Please refer to the table below for a comparison on the 5-class miniImageNet dataset.
>
> |Method|1-shot|5-shot|
> |-|-|-|
> |ABML [1]|$45.0_{\pm 0.6}$|$62.8_{\pm 0.74}$|
> |BMAML [2]|$53.80_{\pm 1.46}$|not provided|
> |PLATIPUS [3]|$50.13_{\pm 1.86}$|not provided|
> |MAML+MetaNCoV (ours)|$\mathbf{57.74}_{\pm 1.47}$|$\mathbf{70.72}_{\pm 0.70}$|

---

> > ### Comment · Reviewer_XQHx · 2024-08-11
> > **Answer**
> >
> > Dear authors,
> >
> > thank you for the additional clarifications and experiments.
> > There are no additional questions from my side. If you include these experiments and the discussion provided in the answers in the paper, I think this will contribute to make the paper more solid and convincing.
> > I'm going to raise my score and I want also to extend my congratulations for the hard work.

---

> > > ### Author Response · Authors · 2024-08-11
> > >
> > > Thank you for your time and insightful suggestions, as well as for raising the score. We will incorporate the above discussions and comparisons into the revised version of our manuscript.

---

### Official Review · Reviewer_BS6h · 2024-07-09

**Soundness:** 2
**Presentation:** 2
**Contribution:** 2
**Rating:** 5
**Confidence:** 3

**Summary:**

This paper proposes a novel approach to meta-learning using normalizing flows for modelling the prior distribution of parameters, with the task solver in the inner loop as the maximum a posterior estimator. Compared with the previous methods, such as MAML, which assume a Gaussian prior, suffering from limited expressiveness, non-injective Sylvester normalizing flows (NFs) are utilised for more flexibility. The method involves propagating samples for each task through the flow and updating the flow based on an average overall task. Experimental results show that this approach consistently outperforms some existing meta-learning techniques.

**Strengths:**

1. This paper is well-written and easy to follow. The motivation for modelling the meta-prior with a very flexible distribution is valid.
2. The background of meta-learning in the few-shot learning is well introduced.

**Weaknesses:**

1. I find the non-injective Sylvester Flows part confusing. In traditional Sylvester Flow models, a specific structure is used to ensure invertibility by imposing conditions on the diagonals of triangular matrices. But in this work, I do not see it.

2. In line 256, the authors claim that while optimization Eq.10b, the latent variables share similarities with LEO, which is confusing. The latent variable in LEO is introduced by its specific architecture and in this submission it refers very different thing. Can the authors explain more about this?

3. [1] handles the prior learning problem for meta-learning from a similar perspective and outperforms MetaNCoV in some settings.

[1] Zhang X, Meng D, Gouk H, Hospedales TM. Shallow bayesian meta learning for real-world few-shot recognition. In Proceedings of the IEEE/CVF international conference on computer vision 2021 (pp. 651-660).

**Questions:**

See the weaknesses section

**Limitations:**

See the weaknesses section

---

> ### Author Rebuttal · Authors · 2024-08-05
>
> Thank you for providing the insightful suggestions. The concerns are addressed one-by-one as follows.
>
> **Regarding Weakness 1**
>
> Our Sylvester NCoV *intentionally* forgoes the injectivity constraint to enhance prior expressiveness, as illustrated in Theorem 3.1. Upon waiving the injectivity, the function $f$ is no longer bijective and thus can be non-invertible. Consequently, the Jacobian $J_f$ needs not to be a positive definite triangular matrix.
>
> **Regarding Weakness 2**
>
> The claim in line 256 aims to illustrate that both MetaNCoV and LEO optimize a latent variable instead of the primal $\boldsymbol{\phi}_t$. We agree with the reviewer that our latent variable is designed from a *different* perspective. We will rewrite this sentence to emphasize this difference and avoid potential confusion.
>
> **Regarding Weakness 3**
>
> Our method distinguishes from [1] in three key aspects:
> - Prior to learn. The proposed MetaNCoV aims to learn the prior $p(\boldsymbol{\phi}_t)$ over *model parameters* $\boldsymbol{\phi}_t$, while [1] propose to learn the prior $p(g(\mathbf{x}_t^n))$ over the extracted *features* $g(\mathbf{x}_t^n)$. Additionally, [1] relies on *preselected* prior form (i.e., conjugate prior) to ensure tractability, whereas our goal is to meta-learn a *data-driven* prior of *universal* forms.
> - Task-level optimization. This work belongs to the optimization-based meta-learning, which finetunes the entire nonlinear model during task-level optimization. In contrast, [1] is a metric-based approach that freezes the feature extractor and adapts merely the last linear layer of the model.
> - Randomness. Our approach is deterministic (cf. maximum a posteriori (10b)), focusing on enhanced prior expressiveness without necessitating a tractable pdf. In comparison, [1]'s task-level optimization seeks the probabilistic posterior distribution, which concentrates on uncertainty quantification and pdf tractability.
>
> Therefore, [1] is parallel to our work and not directly comparable. Thanks for pointing out this related work; these differences will be highlighted in Appendix E of the revised paper.
>
>
> [1] X. Zhang, D. Meng, H. Gouk, and T. Hospedales, "Shallow Bayesian meta learning for real-world few-shot recognition," ICCV 2021.

---

> ### Comment · Reviewer_BS6h · 2024-08-13
>
> I thank the authors for their efforts in addressing my concerns in terms of weaknesses 1 and 2. I still hold different ideas about the comparison with [1] but I will change my score to borderline accept.

---

> > ### Author Response · Authors · 2024-08-13
> >
> > Thank you for your constructive questions and suggestions. We will incorporate the points discussed above into our revised manuscript.

---

### Official Review · Reviewer_4d6S · 2024-07-10

**Soundness:** 3
**Presentation:** 3
**Contribution:** 3
**Rating:** 6
**Confidence:** 5

**Summary:**

One of representative optimization-based meta-learning method is Model-agnostic meta-learning (MAML), where the inner loop can be interpreted as solving MAP. Here, the prior distribution over model parameters is defined as Gaussian pdf and the shared initialization is the mean parameter of the Gaussian pdf. The choice of Gaussian distribution for prior distribution may lack expressiveness; therefore, this paper proposes a new meta-learning method by introducing non-injective change-of-variable (NCoV) model for prior distribution over task parameters. Furthermore, the proposed method does not need to meta-learn shared initialization as done in MAML by feed-forwarding the zero vector into the normalizing flow (choosing the base distribution of normalizing flow as standard Gaussian). The empirical results shows the efficacy of proposed method in few-shot learning settings.

**Strengths:**

- This paper propose a principled meta-learning method to improve expressiveness of prior distribution over model parameters, by theoritically derived method for non-injective change-of-variable models.

- The proposed method is more efficient when the dimensionality of parameter increases, while the pre-defined prior distribution baselines is not scalable to the dimensionality of parameters.

- The empiricial results are very strong.

**Weaknesses:**

- This paper does not provide the number of "meta-"parameters for each method in the experiment section. I wonder whether the improvement comes from the expressiveness of proposed method or the sufficient number of parameters $\theta_f$ for modeling normalizing flow.

- This paper does not discuss relevant literature in main paper (does not include related work section). Especially, I am curious about the novelty of NCoV models derived in this paper in comparison to the literature of normalizing flows. If NCoV models have already been studied in the literature, this paper seems to lack novelty. If not, I would like to recommend the authors to include related work or discussion to highlight their contribution with respect to normalizing flows.

- This paper follows experimental setups for the convention of meta-learning, but the empircial validation is conducted in small-scale settings. This paper argue that the proposed method is more scalable to dimensionality of model parameters, therefore, it would be better to highlight the argument by conducting experiments on more larger models (e.g., VIT).

**Questions:**

Please see weakness section.

**Limitations:**

This paper discuss limitation in the conclusion section.

---

> ### Author Rebuttal · Authors · 2024-08-05
>
> Thanks for the valuable feedback provided. The questions raised are addressed one-by-one below.
>
> **Regarding weaknesses 1 and 2**
>
> We would like to kindly bring to the reviewer's attention that the model complexity with meta-parameter dimension $D$ has been already studied extensively in Appendix D.4; additionally, related works have been provided in Appendix E of the original submission. Moreover, we have referred to them in lines 327 and 193 of the main paper to guide the readers to relevant Appendices. As evident from Table 6 in Appendix D.4, parameters of our MetaNCoV is notably less than MC, and its time- and space-complexity is comparable to popular optimization-based approaches. Furthermore, while NFs are thoroughly studied, the NCoV models have gained little attention in machine learning and statistical learning due to their intractability; see lines 701-706 in Appendix E for related works and corresponding comparisons.
>
>
> **Regarding weakness 3**
>
> Our main claim in this work is that MetaNCoV empowers more *expressive* (rather than *scalable*) prior in high-dimensional spaces; cf. lines 12, 64, 176, and 188. This argument has been also empirically confirmed in our numerical tests in Section 4 and Appendix D. In terms of scalability, MetaNCoV is comparable to popular optimization-based approaches such as MAML and MC; see Table 6. However, it is still challenging to directly apply these algorithms (even vanilla MAML) to extremely large-scale models such as vision transformers. Potential solutions include restricting the task-level update (1b) to only a *subset* of parameters, or resorting to corresponding first-order variants such as FO-MAML.

---

> ### Comment · Reviewer_4d6S · 2024-08-13
> **Answer**
>
> Dear authors
>
> I thank the authors for the effort in clarifying my concerns. I will change my score to weak accept.

---

> > ### Author Response · Authors · 2024-08-13
> >
> > Thank you for your valuable time and insightful questions, as well as for updating the rating.

---

### Official Review · Reviewer_VcV4 · 2024-07-15

**Soundness:** 3
**Presentation:** 3
**Contribution:** 2
**Rating:** 5
**Confidence:** 3

**Summary:**

This paper proposes to use non-injective change of variables theorem to make the prior in meta-learning more flexible than fixed shaped ones that have previously been used. Abundent theoretical analysis and experimental results support their claim that more flexible meta-level prior pdf can significantly boost the performance of meta-learning in a few simple benchmark settings.

**Strengths:**

- The paper is well written and easy to understand
- Theoretical analysis is done rigorously
- Experimental results are strong (but limited to too outdated benchmarks)

**Weaknesses:**

Too limited benchmarks. While the idea and the theoretical analysis are nice, the benchmarks used in this paper -- mini-Imagenet, tieredImageNet, and CUB are too outdated in my opinion. They were popular in 5 years ago, and nowadays people are actually no longer interested in these benchmarks, especially in the LLM era. I think the minimum level should be meta-dataset. Could you provide the results on meta-dataset?

**Questions:**

I wonder about the positioning of this paper. Does the proposed method have to be specifically tailored to meta-learning only? Why don't you broaden the scope of the paper by doing experiments on more standard non meta-learning setup (like in the synthetic experiments)?

**Limitations:**

The authors partially addressed the limitations of their work.

---

> ### Author Rebuttal · Authors · 2024-08-05
>
> Thank you for the interest in this work and the constructive feedback provided, which have been carefully addressed as follows.
>
> **Response to weaknesses**
>
> Due to limited time and computational resources, we are unfortunately unable to provide the results on the full MetaDataset that contains 240GB data. However, it is feasible to conduct a similar experiment on a smaller scale.
>
> Before providing the details, we would like to emphasize that MetaDataset aims to assess the *cross-domain generalization* performance of meta-learning algorithms, which comprises 10 datasets from various domains, including ImageNet, Aircraft, and CUB. The standard setup involves meta-training a model on ImageNet (or the entire MetaDataset), and then meta-testing the trained model on all datasets.
>
> Following the cross-domain few-shot learning setup in [1], we meta-train our prior model on miniImageNet, and meta-test it on tieredImageNet, Cars, and CUB datasets. As miniImageNet is a subset of the full ImageNet, we believe this test to some extent reflects the promising cross-domain generalization performance of our proposed algorithm. As shown in the Table below, our method consistently outperforms popular meta-learning approaches in such a setup, especially in the 1-shot case. This not only confirms the cross-domain generalization of MetaNCoV, but again justifies the importance of expressive prior when data are exceedingly limited.
>
> |Method|5-way|TieredImageNet|5-way|CUB|5-way|Cars|
> |-|:-:|:-:|:-:|:-:|:-:|:-:|
> | |1-shot|5-shot|1-shot|5-shot|1-shot|5-shot|
> |MAML [2]|$51.61_{\pm 0.20}$|$65.76_{\pm 0.27}$|$40.51_{\pm 0.08}$|$53.09_{\pm 0.16}$|$33.57_{\pm 0.14}$|$44.56_{\pm 0.21}$|
> |ANIL [1]|$52.82_{\pm 0.29}$|$66.52_{\pm 0.28}$|$41.12_{\pm 0.15}$|$55.82_{\pm 0.21}$|$34.77_{\pm 0.31}$|$46.55_{\pm 0.29}$|
> |BOIL [3]|$53.23_{\pm 0.41}$|$69.37_{\pm 0.23}$|$44.20_{\pm 0.15}$|$60.92_{\pm 0.11}$|$36.12_{\pm 0.29}$|$50.64_{\pm 0.22}$|
> |Sparse-MAML+ [4]|$53.91_{\pm 0.67}$|$69.92_{\pm 0.21}$|$43.43_{\pm 1.04}$|$62.02_{\pm 0.78}$|$37.14_{\pm 0.77}$|$53.18_{\pm 0.44}$|
> |GAP [5]|$58.56_{\pm 0.93}$|$72.82_{\pm 0.77}$|$44.74_{\pm 0.75}$|$64.88_{\pm 0.72}$|$38.44_{\pm 0.77}$|$55.04_{\pm 0.77}$|
> MetaNCoV (ours)|$\mathbf{61.50}_{\pm 1.49}$|$\mathbf{73.10}_{\pm 0.74}$|$\mathbf{47.84}_{\pm 1.49}$|$\mathbf{65.27}_{\pm 0.73}$|$\mathbf{41.66}_{\pm 1.48}$|$\mathbf{57.19}_{\pm 0.75}$|
>
> [1] A. Raghu, M. Raghu, S. Bengio, and O. Vinyals, "Rapid learning or feature reuse? towards understanding the effectiveness of MAML," ICLR 2020.
>
> [2] C. Finn, P. Abbeel, and S. Levine, "Model-agnostic meta-learning for fast adaptation of deep networks," ICML 2017.
>
> [3] J. Oh, H. Yoo, C. Kim, and S.-Y. Yun, "Boil: Towards representation change for few-shot learning," ICLR 2021.
>
> [4] J. von Oswald, D. Zhao, S. Kobayashi, S. Schug, M. Caccia, N. Zucchet, and J. a. Sacramento, "Learning where to learn: Gradient sparsity in meta and continual learning," NeurIPS 2021.
>
> [5] S. Kang, D. Hwang, M. Eo, T. Kim, and W. Rhee, "Meta-learning with a geometry-adaptive preconditioner," CVPR, 2023.
>
> **Response to questions**
>
> While our idea has the potential to be broadened beyond meta-learning, we must emphasize that our current setup is specifically tailored to meta-learning, which does not require a tractable pdf, but rather demands enhanced prior expressiveness. We should also highlight to the reviewer that the *intractability* of pdf prohibits learning NCoV via conventional approaches such as maximum likelihood training and evidence lower-bound maximization -- this thus necessitates careful attention and extra certain designs when applying our idea to other domains.

---

### Decision · Program_Chairs · 2024-09-25

**Decision:**

Accept (poster)

**Comment:**

**Summary of the paper:**
The paper advocates for learning a flexible (non-prescribed) prior in meta learning and achieves this by the change of variable trick from a fixed prior; differently to NFs, however, without requiring the invertibility. Particularly, it works on optimization-based meta learning and uses Sylvester NF without enforcing the Jacobian structure. It evaluates the method on several few-shot learning benchmarks of meta learning.

**Summary of the reviews:**
The reviewers found the paper clearly written and well-motivated, the theoretical and empirical results rigorously achieved, and the empirical results to be significantly improving over the baselines with fixed form of priors.

On the other hand, they were concerned about limited benchmarks and novelty over works within Bayesian (or generally probabilistic) meta learning.

**Summary of the rebuttal and discussions:**
In the rebuttal period, the authors provided additional results on a lower-scale version of MetaDataset for cross-domain generalization where the trends are seen to be similar. They also argue for the MAP aspect of their work when optimizing with the learned prior on test tasks which trades-off the uncertainty quantification of Bayesian meta learning for a tractable expressivity of their learnt priors.

**Consolidation report:**
The authors successfully addressed the main criticisms of the reviewers that led to their eventual leaning towards acceptance.

**Recommendation:**
The AC agrees with the unanimous rating and suggests acceptance.